# TEAC: Integrating Trust Region and Max Entropy Actor Critic for Continuous Control

## Abstract

Trust region methods and maximum entropy methods are two state-of-the-art branches used in reinforcement learning (RL) for the benefits of stability and exploration in continuous environments, respectively. This paper proposes to integrate both branches in a unified framework, thus benefiting from both sides. We first transform the original RL objective to a constraint optimization problem and then proposes trust entropy actor-critic (TEAC), an off-policy algorithm to learn stable and sufficiently explored policies for continuous states and actions. TEAC trains the critic by minimizing the refined Bellman error and updates the actor by minimizing KL-divergence loss derived from the closed-form solution to the Lagrangian. We prove that the policy evaluation and policy improvement in TEAC is guaranteed to converge. We compare TEAC with 4 state-of-the-art solutions on 6 tasks in the MuJoCo environment. The results show that TEAC with optimized parameters achieves similar performance in half of the tasks and notably improvement in the others in terms of efficiency and effectiveness.

## 1 Introduction

With the use of high-capacity function approximators, such as neural networks, reinforcement learning (RL) becomes practical in a wide range of real-world applications, including game playing (Mnih et al., 2013; Silver et al., 2016) and robotic control (Levine et al., 2016; Haarnoja et al., 2018a). However, when dealing with the environments with continuous state space or/and continuous action space, most existing deep reinforcement learning (DRL) algorithms still suffer from unstable learning processes and are impeded from converging to the optimal policy.

The reason for unstable training process can be traced back to the use of greedy or $\epsilon$-greedy policy updates in most algorithms. With the greedy update, a small error in value functions may lead to abrupt policy changes during the learning iterations. Unfortunately, the lack of stability in the training process makes the DRL unpractical for many real-world tasks (Peters et al., 2010; Schulman et al., 2015; Tangkaratt et al., 2018). Therefore, many policy-based methods have been proposed to improve the stability of policy improvement (Kakade, 2002; Peters & Schaal, 2008; Schulman et al., 2015; 2017). Kakade (2002) proposed a natural policy gradient-based method which inspired the design of trust region policy optimization (TRPO). The trust region, defined by a bound of the Kullback-Leibler (KL) divergence between the new and old policy, was formally introduced in Schulman et al. (2015) to constrain the natural gradient policy changing within the field of trust. An alternative to enforcing a KL divergence constraint is to utilize the clipped surrogate objective, which was used in Proximal Policy Optimization (PPO) (Schulman et al., 2017) to simplify the objective of TRPO while maintaining similar performance. TRPO and PPO have shown significant performance improvement on a set of benchmark tasks. However, these methods are all on-policy methods requiring a large number of on-policy interaction with environment for each gradient step. Besides, these methods focus more on the policy update than exploration, which is not conducive to finding the global optimal policy.

The globally optimal behavior is known to be difficult to learn due to sparse rewards and insufficient explorations. In addition to simply maximize the expected reward, maximum entropy RL (MERL) (Ziebart et al., 2008; Toussaint, 2009; Haarnoja et al., 2017; Levine, 2018) proposes to extend the conventional RL objective with an additional "entropy bonus" argument, resulting in the preferences to the policies with higher entropy. The high entropy of the policy explicitly encourages

exploration, thus improving the diverse collection of transition pairs, allowing the policy to capture multi-modes of good policies, and preventing from premature convergence to local optima. MERL reforms the reinforcement learning problem into a probabilistic framework to learn energy-based policies to maintain the stochastic property and seek the global optimum. The most representative methods in this category are soft Q-learning (SQL) (Haarnoja et al., 2017) and Soft Actor Critic (SAC) (Haarnoja et al., 2018b;c). SQL defines a soft Bellman equation and implements it in a practical off-policy algorithm which incorporates the entropy of the policy into the reward to encourage exploration. However, the actor network in SQL is treated as an approximate sampler, and the convergence of the method depends on how well the actor network approximates the true posterior. To address this issue, SAC extends soft Q-learning to actor-critic architecture and proves that a given policy class can converge to the optimal policy in the maximum entropy framework. However, off-policy DRL is difficult to stabilize in policy improvement procedure (Sutton & Barto, 1998; van Hasselt et al., 2018; Ciosek et al., 2019) which may lead to catastrophic actions, such as ending the episode and preventing further learning.

Several models have been proposed to benefit from considering both the trust region constraint and the entropy constraint, such as MOTO (Akrour et al., 2016) , GAC (Tangkaratt et al., 2018), and Trust-PCL (Nachum et al., 2018). However, MOTO and GAC cannot efficiently deal with high-dimensional action space because they rely on second-order computation, and Trust-PCL suffers from algorithm efficiency due to its requirement of trajectory/sub-trajectory samples to satisfy the pathwise soft consistency.

Therefore, in this paper, we propose to further explore the research lines of unifying trust region policy-based methods and maximum entropy methods. Specifically, we first transform the RL problem into a primal optimization problem with four additional constraints to 1) set an upper bound of KL divergence between the new policy and the old policy to ensure the policy changes are within the region of trust, 2) provide a lower bound of the policy entropy to prevent from a premature convergence and encourage sufficient exploration, and 3) restrain the optimization problem as a Markov Decision Process (MDP). We then leverage the Lagrangian duality to the optimization problem to redefine the Bellman equation which is used to verify the policy evaluation and guarantee the policy improvement. Thereafter, we propose a practical trust entropy actor critic (TEAC) algorithm, which trains the critic by minimizing the refined Bellman error and updates the actor by minimizing KL-divergence loss derived from the closed-form solution to the Lagrangian. The update procedure of the actor involves two dual variables w.r.t. the KL constraint and entropy constraint in the Lagrangian. Based on the Lagrange dual form of the primal optimization problem, we develop gradient-based method to regulate the dual variables regarding the optimization constraints.

The key contribution of the paper is a novel off-policy trust-entropy actor-critic (TEAC) algorithm for continuous controls in DRL. In comparison with existing methods, the actor of TEAC updates the policy with the information from the old policy and the exponential of the current Q function, and the critic of TEAC updates the Q function with the new Bellman equation. Moreover, we prove that the policy evaluation and policy improvement in trust entropy framework is guaranteed to converge. A detailed comparison with similar work, including MOTO (Akrour et al., 2016), GAC (Tangkaratt et al., 2018), and Trust-PCL (Nachum et al., 2018), is provided in Sec. 4 to explain that TEAC is the most effective and most theoretically complete method. We compare TEAC with 4 state-of-the-art solutions on the tasks in the MuJoCo environment. The results show that TEAC is comparable with the state-of-the-art solutions regarding the stability and sufficient exploration.

## 2 PRELIMINARIES

A RL problem can be modeled as a standard Markov decision process (MDP), which is represented as a tuple $\langle \mathcal{S}, \mathcal{A}, r, p, p_0, \gamma \rangle$. $\mathcal{S}$ and $\mathcal{A}$ denote the state space and the action space, respectively. $p_0(s)$ denotes the initial state distribution. At time $t$, the agent in state $s_t$ selects an action $a_t$ according to the policy $\pi(a|s)$, in which the performance of the *state-action* pair is quantified by the reward function $r(s_t, a_t)$ and the next state of the agent is decided by the transition probability as $s_{t+1} \sim p(s_{t+1}|s_t, a_t)$. The goal of the agent is to find the optimal policy $\pi(a|s)$ to maximize the expected reward $\mathbb{E}_{s_0, a_0, \dots} [\sum_{t=0}^{\infty} \gamma^t r(s_t, a_t)]$, where $s_0 \sim p_0(s)$ and $s_{t+1} \sim p(s_{t+1}|s_t, a_t)$. $\gamma$ is a discount factor $(0 < \gamma < 1)$) which quantifies how much importance we give for future rewards.

The state-action value function $Q^\pi(s_t, a_t)$ and the value function $V^\pi(s_t)$ are then defined as:

$$Q^\pi(s_t, a_t) = \mathbb{E}_{s_{t+1}, a_{t+1}, \ldots}\left[\sum_{l=0}^{\infty} \gamma^l r(s_{t+l}, a_{t+l})\right], V^\pi(s_t) = \mathbb{E}_{a_t, s_{t+1}, \ldots}\left[\sum_{l=0}^{\infty} \gamma^l r(s_{t+l}, a_{t+l})\right].$$

For the continuous environments, which is the focus of this paper, $\mathcal{S}$ and $\mathcal{A}$ denote finite dimensional real valued vector spaces, $s$ denotes the real-valued state vector, and $a$ denotes the real-valued action vector. The expected reward can be defined as :

$$\mathcal{J}(\pi) = \mathbb{E}_{(s,a)\sim\rho_\pi(s,a)}\left[Q^\pi(s, a)\right] = \mathbb{E}_{\rho_\pi(s)\pi(a|s)}\left[Q^\pi(s, a)\right], \tag{1}$$

where $\rho_\pi(s)$ and $\rho_\pi(s, a)$ denote the (discounted) state and (discounted) state-action marginals of the trajectory distribution induced by a policy $\pi(a|s)$. [1]

## 3 OUR METHOD

This section explains the details and features of the TEAC framework with the focus on the mathematical deductions and proofs of the guaranteed policy improvement and convergence in an actor-critic architecture.

### 3.1 PRIMAL AND DUAL OPTIMIZATION PROBLEM

To stabilize the training process and steer the exploration, in addition to simply maximizing the expected reward with ($\epsilon$-) greedy policy updates, we propose to 1) confine the KL-divergence between neighboring policies in the training procedure to avoid large-step policy updates, and 2) favor a stochastic policy with relatively larger entropy to avoid premature convergence due to insufficient exploration. Therefore, we define the RL problem as a primal optimization problem with additional constraints, given as:

$$
\begin{aligned}
\max_\pi \quad & \mathbb{E}_{\rho_\pi(s)\pi(a|s)}[\hat{Q}(s, a)], \\
\text{subject to} \quad & \mathbb{E}_{\rho_\pi(s)}\left[\text{KL}\left(\pi(\cdot|s)\|\pi_{\text{old}}(\cdot|s)\right)\right] \leq \tau, \\
& \mathbb{E}_{\rho_\pi(s)}[\text{H}(\pi(\cdot|s))] \geq \eta, \\
& \mathbb{E}_{\rho_\pi(s)}\int \pi(a|s)da = 1, \\
& \mathbb{E}_{\rho_\pi(s)\pi(a|s)p(s'|s,a)}\hat{V}(s') = \mathbb{E}_{\rho_\pi(s')}\hat{V}(s'),
\end{aligned}
\tag{2}
$$

where $\hat{Q}(s, a)$ is a critic estimating the state-action value function whose parameter is learned such that $\hat{Q}(s, a) \approx Q^\pi(s, a)$, $\pi(\cdot|s)$ is the policy distribution to be learned, $\pi_{\text{old}}(\cdot|s)$ is the prior policy distribution, and $\hat{V}(s')$ is a state feature function estimating the state value function of the next state. The term $\text{KL}\left(\pi(\cdot|s)\|\pi_{\text{old}}(\cdot|s)\right) = \mathbb{E}_{\pi(a|s)}[\log \pi(a|s) - \log \pi_{\text{old}}(a|s)]$ confines the KL-divergence between the distributions of the new and old policies. The third constraint ensures that the state-action marginal of the trajectory distribution is a proper probability density function. As the state marginal of the trajectory distribution needs to comply with the policy $\pi(a|s)$ and the system dynamics $p(s'|s, a)$, i.e., $\rho_\pi(s)\pi(a|s)p(s'|s, a) = \rho_\pi(s')$, meanwhile the direct matching of the state probabilities is not feasible in continuous state spaces, the use of $\hat{V}(s')$ in the fourth constraint which can be also considered as state features, helps to focus on matching the feature averages. These last two constraints formally restrain the optimization problem within a MDP framework.

The objective is to maximize the expected reward of a policy while ensuring it satisfies the lower bound of entropy and upper bound of distance from the previous policy. The constraint of KL-divergence term helps to avoid the abrupt difference between the new and old policies, while the constraint of the entropy term helps to promote the policy exploration.

The entropy constraint is crucial in our optimization problem for two reasons: 1) Prior studies show that the use of KL-bound leads to a rapid decrease of the entropy, thus bounding the entropy helps to lower the risk of premature convergence induced by the KL-bound; 2) Each iteration of policy update will modify the critic $\hat{Q}(s, a)$ and the state distribution $\rho_\pi(s)$, thus changing the optimization

---

[1] Following Sutton et al. (2000), we use $\rho_\pi$ in the paper to implicate that $\rho_\pi$ is the stationary distribution of states under $\pi$ and independent of $s_0$ for all policies.

landscape of the policy parameters. The entropy constraint ensures the exploration in the action space in case of evolving optimization landscapes.

The Lagrangian of this optimization problem is denoted as:

$$
\begin{aligned}
\mathcal{L}(\pi, \alpha, \beta, \lambda, \nu) =& \mathbb{E}_{\rho(\boldsymbol{s})\pi(\boldsymbol{a}|\boldsymbol{s})}[\hat{Q}(\boldsymbol{s}, \boldsymbol{a})] + \alpha\left(\tau - \mathbb{E}_{\rho(\boldsymbol{s})}\left[\text{KL}\left(\pi(\cdot|\boldsymbol{s})\|\pi_{\text{old}}(\cdot|\boldsymbol{s})\right)\right]\right) \\
& + \beta\left(\mathbb{E}_{\rho(\boldsymbol{s})}[\text{H}(\pi(\cdot|\boldsymbol{s}))] - \eta\right) + \lambda\left(\mathbb{E}_{\rho(\boldsymbol{s})}\int \pi(\boldsymbol{a}|\boldsymbol{s})d\boldsymbol{a} - 1\right) \\
& + \nu(\mathbb{E}_{\rho(\boldsymbol{s})\pi(\boldsymbol{a}|\boldsymbol{s})p(\boldsymbol{s}'|\boldsymbol{s},\boldsymbol{a})}\hat{V}(\boldsymbol{s}') - \mathbb{E}_{\rho(\boldsymbol{s}')}\hat{V}(\boldsymbol{s}')),
\end{aligned}
\tag{3}
$$

where $\alpha$, $\beta$, $\lambda$, $\nu$ are the dual variables, and for the sake of brevity, we use $\rho(\boldsymbol{s})$ to represent $\rho_\pi(\boldsymbol{s})$. Eq. 3 is a super set of trust region and maximum entropy methods. That is, $\beta = 0$ leads to an equivalent objective function as the standard trust region, while $\alpha = 0$, which indicates that the KL-divergence bound is not active, leads to a maximum entropy RL objective that SAC tries to solve.

Take derivative of $\mathcal{L}$ w.r.t. $\pi$ and set the derivative to zero:

$$
\begin{aligned}
\partial_\pi \mathcal{L} =& \mathbb{E}_{\rho(\boldsymbol{s})}\Bigg[\int\bigg(\hat{Q}(\boldsymbol{s}, \boldsymbol{a}) - (\alpha + \beta)\log\pi(\boldsymbol{a}|\boldsymbol{s}) + \alpha\log\pi_{\text{old}}(\boldsymbol{a}|\boldsymbol{s}) - \nu\hat{V}(\boldsymbol{s}) + \\
& \mathbb{E}_{p(\boldsymbol{s}'|\boldsymbol{s},\boldsymbol{a})}[\nu\hat{V}(\boldsymbol{s}')]\bigg)d\boldsymbol{a}\Bigg] - (\alpha + \beta + \lambda) \\
=& \hat{Q}(\boldsymbol{s}, \boldsymbol{a}) - (\alpha + \beta)\log\pi(\boldsymbol{a}|\boldsymbol{s}) + \alpha\log\pi_{\text{old}}(\boldsymbol{a}|\boldsymbol{s}) - \nu\hat{V}(\boldsymbol{s}) + \mathbb{E}_{p(\boldsymbol{s}'|\boldsymbol{s},\boldsymbol{a})}[\nu\hat{V}(\boldsymbol{s}')] \\
& - (\alpha + \beta + \lambda) \\
=& 0.
\end{aligned}
\tag{4}
$$

Continuous problem domains require a practical approximation to the policy update function. We use neural networks as function approximators to parameterize the policy and Q function. Specifically, the Q function, known as critic, is modeled as expressive neural networks $Q_\phi(\boldsymbol{s}, \boldsymbol{a})$, and we follow Lillicrap et al. (2016) to build a target critic network $Q_{\bar{\phi}}$ which mitigates the challenge of overestimation. Meanwhile, the policy, known as actor, is parameterized by $\pi_\theta(\cdot|\boldsymbol{s})$ as a Gaussian with mean and covariance given by neural networks, and we also build up another neural network $\pi_{\hat{\theta}}(\cdot|\boldsymbol{s})$ with the same architecture as $\pi_\theta$ to enable us to facilitate policy learning by leveraging the "old" policy within our framework.

## 3.2 CRITIC UPDATE

Given the fact that we sample actions from the actor network as parameterized Gaussian distribution and the value function should satisfy Bellman equation[2],

$$
\hat{V}(\boldsymbol{s}) = \hat{Q}(\boldsymbol{s}, \boldsymbol{a}) - (\alpha + \beta)\log\pi(\boldsymbol{a}|\boldsymbol{s}) + \alpha\log\pi_{\text{old}}(\boldsymbol{a}|\boldsymbol{s}) + \mathbb{E}_{p(\boldsymbol{s}'|\boldsymbol{s},\boldsymbol{a})}[\hat{V}(\boldsymbol{s}')] - (\alpha + \beta + \lambda) \tag{5}
$$

The last constant term in Eq.5 can be ignored as it does not affect the Bellman iteration when neural networks are used to approximate the value function. Therefore, the Bellman equation can be redefined in our framework.

According to Eq.5, we could compute the value of a fixed policy $\pi$. Starting from any function $Q : S \times A \to \mathbb{R}$, we define our modified Bellman backup operator.

**Definition 3.1** *Bellman Equation. A modified Bellman backup operator $\mathcal{T}^\pi$ is defined as*

$$
\mathcal{T}^\pi Q(\boldsymbol{s}, \boldsymbol{a}) \triangleq r + \gamma\mathbb{E}_{p(\boldsymbol{s}'|\boldsymbol{s},\boldsymbol{a})}[V(\boldsymbol{s}')], \tag{6}
$$

*where*

$$
V(\boldsymbol{s}) = \mathbb{E}_{\pi(\boldsymbol{a}|\boldsymbol{s})}\left[Q(\boldsymbol{s}, \boldsymbol{a}) - (\alpha + \beta)\log\pi(\boldsymbol{a}|\boldsymbol{s}) + \alpha\log\pi_{old}(\boldsymbol{a}|\boldsymbol{s})\right] \tag{7}
$$

*is the trust entropy state-value function in our framework.*

---

[2] we could utilize any form of state features $\hat{V}(\boldsymbol{s}')$. Thus, $\nu\hat{V}(\boldsymbol{s}')$ can be seen as another form of state features. Therefore, $\nu$ can be arbitrary

In the sequel, $Q(s, a)$ stands for the state-action value function obtained by iteratively applying the modified Bellman backup operator in Eq.6 and Eq.7, which is the trust entropy Q-value in our framework. Meanwhile, the policy evaluation can be verified accordingly.

**Lemma 1** *(Trust Entropy Policy Evaluation). Let $Q^{k+1} = \mathcal{T}^\pi Q^k$, the sequence $Q^k$ will converge to the trust entropy Q-value of $\pi$ as $k \to \infty$ when considering a mapping $Q^0 : \mathcal{S} \times \mathcal{A} \to \mathbb{R}$ with $|\mathcal{A}| < \infty$ and the Bellman backup operator $\mathcal{T}^\pi$.*

*Proof.* See Appendix A.1.

The learning of Q function can utilize off-policy methods to acquire high sample efficiency. Hence, the parameters can be trained by minimizing the squared Bellman error, given as:

$$\mathcal{L}_Q(\phi) = \mathbb{E}_{(s,a)\sim\mathcal{D}} \left[ \frac{1}{2} \left( Q_\phi(s,a) - y \right)^2 \right], \tag{8}$$

where $\mathcal{D}$ is the replay buffer, $Q_\phi(s, a)$ represents the Q network (also known as critic network) which is parameterized by $\phi$, and $y = r(s, a) + \gamma \mathbb{E}_{s'\sim p} \left[ V_{\bar{\phi}}(s') \right]$, where $V_{\bar{\phi}}$ denotes the value function obtained from the target critic network $Q_{\bar{\phi}}$ with Eq. 7. The inclusion of the target critic network helps to stabilize the training. As suggested in Lillicrap et al. (2016), $\bar{\phi}$ is updated via $\bar{\phi} \leftarrow \kappa\phi + (1 - \kappa)\bar{\phi}$ where $0 \leq \kappa \leq 1$. We set $\kappa = 0.005$ in the experiments. The expected squared Bellman error is computed with samples drawn from the replay buffer using mini-batch. Thus, the approximate gradient of the squared Bellman error $\mathcal{L}_Q(\phi)$ w.r.t. $\phi$ is:

$$\begin{aligned}
\hat{\nabla}_\phi \mathcal{L}_Q(\phi) =& \nabla_\phi Q_\phi(a, s) \left( Q_\phi(s,a) - \left( r(s,a) + \gamma \left( Q_{\bar{\phi}}(s',a') - \right.\right.\right. \\
& \left.\left.\left. (\alpha + \beta) \log \pi_\theta(a|s') + \alpha \log \pi_{\hat{\theta}}(a|s') \right) \right) \right),
\end{aligned} \tag{9}$$

where $\theta$ and $\hat{\theta}$ are parameters of current policy and old policy respectively, and $a'$ is sampled from current policy $\pi_\theta$ given $s'$.

### 3.3 ACTOR UPDATE

Setting $\nu = 0$ in Eq.4 in fact does not change the optimization problem. Therefore, a closed-form solution regarding the policy is given as:

$$\begin{aligned}
\pi(a|s) &= \pi_{\text{old}}(a|s)^{\frac{\alpha}{\alpha+\beta}} \exp \left( \frac{\hat{Q}(s,a)}{\alpha+\beta} \right) \exp \left( -\frac{\alpha+\beta+\lambda}{\alpha+\beta} \right) \\
&\propto \pi_{\text{old}}(a|s)^{\frac{\alpha}{\alpha+\beta}} \exp \left( \frac{\hat{Q}(s,a)}{\alpha+\beta} \right),
\end{aligned} \tag{10}$$

where $\exp \left( -\frac{\alpha+\beta+\lambda}{\alpha+\beta} \right)$ is the normalization term of $\pi(a|s)$ (The detailed derivation is provided in A.2). It should be noted that MORE (Daniel et al., 2016), MOTO (Akrour et al., 2016), GAC (Tangkaratt et al., 2018), and Trust-PCL (Nachum et al., 2018) can also been viewed as prior work stemming from Eq.10. However, it is infeasible to use Eq.10 to directly update the policy given the fact that we cannot guarantee that the resulting policy remains in the same policy class when weighing the old policy with the exponential of Q function without any assumption. Different strategies have been applied in the prior work to address this issue, and the detailed discussion is provided in Sec. 4.

To improve the tractability of policies, as Haarnoja et al. (2018c), we require the policy is selected from a set of policies $\pi \in \Pi$, which is a parameterized Gaussian distribution family. This is guaranteed by the use of the Kullback-Leibler divergence to ensure the improved policy locates in the same policy set. Since the normalization term $\exp \left( -\frac{\alpha+\beta+\lambda}{\alpha+\beta} \right)$ is intractable and does not contribute to the gradient of the new policy, it can be ignored. Therefore, the policy is updated by

$$\mathcal{L}_\pi(\theta) = \mathbb{E}_{s\sim\mathcal{D}} \left[ \mathrm{D}_{\mathrm{KL}} \left( \pi_\theta(a|s) \,\|\, \left( \pi_{\hat{\theta}}(a|s)^{\frac{\alpha}{\alpha+\beta}} \exp \left( \frac{Q(s,a)}{\alpha+\beta} \right) \right) \right) \right], \tag{11}$$

where $\mathcal{D}$ is the replay buffer, $\pi_\theta$ represents the parameterized policy, $\pi_{\hat{\theta}}$ represents the parameterized old policy, and $\boldsymbol{a}$ in $Q$ is sampled from the current policy $\pi_\theta$. In practice, the old policy network equals to the policy network in the last iteration. Therefore, we could leverage another actor network to keep the old policy by copying $\theta$ to $\hat{\theta}$ after computing the loss function of the policy and before the back propagation in each iteration (The detailed algorithm is provided in Appendix C). With the assumption of policy being Gaussian, the policy improvement can be guaranteed in our framework.

**Lemma 2** *(Trust Entropy Policy Improvement). Given a policy $\pi$ and an old policy $\hat{\pi}$, define a new policy*

$$\tilde{\pi}(\boldsymbol{a}|\boldsymbol{s}) \propto \hat{\pi}(\boldsymbol{a}|\boldsymbol{s})^{\frac{\alpha}{\alpha+\beta}} \exp\left(\frac{Q^\pi(\boldsymbol{s},\boldsymbol{a})}{\alpha+\beta}\right), \quad \forall \boldsymbol{s}. \tag{12}$$

*If $Q$ is bounded, $\int \hat{\pi}(\boldsymbol{a}|\boldsymbol{s})^{\frac{\alpha}{\alpha+\beta}} \exp\left(\frac{Q^\pi(\boldsymbol{s},\boldsymbol{a})}{\alpha+\beta}\right) \mathrm{d}\boldsymbol{a}$ is bounded for any $\boldsymbol{s}$ (for all $\pi$, $\hat{\pi}$ and $\tilde{\pi}$), and the policies are the parameterized Gaussian networks. Then we can obtain $Q^{\tilde{\pi}}(\boldsymbol{s},\boldsymbol{a}) \geq Q^\pi(\boldsymbol{s},\boldsymbol{a}), \forall \boldsymbol{s},\boldsymbol{a}.$*

*Proof.* See Appendix A.3.

In other words, in the policy improvement, we use the information from the old policy and exponential of the Q function induced by the current policy to derive the policy of next iteration. Because the Q function is a non-linear function approximation parameterized by neural networks and can be differentiated, the reparameterization trick $\boldsymbol{a} = f_\theta(\xi; \boldsymbol{s})$, where $\xi_t$ is an input noise vector sampled from standard normal distribution, can be applied. Then, the approximate gradient of $\mathcal{L}_\pi(\theta)$ w.r.t. $\theta$ is given as:

$$\hat{\nabla}_\theta \mathcal{L}_\pi(\theta) = \nabla_\theta(\alpha+\beta)\log\left(\pi_\theta(\boldsymbol{a}|\boldsymbol{s})\right) - \nabla_{\boldsymbol{a}} Q_\phi(\boldsymbol{s},\boldsymbol{a})\nabla_\theta f_\theta(\xi; \boldsymbol{s}). \tag{13}$$

### 3.4 DUAL VARIABLES UPDATE

This section explains the updates of dual variables ($\alpha$ and $\beta$) throughout the entire framework.

The dual function, which is derived by substituting $\pi(a|s)$ in the Lagrangian (Eq. 3) with its form in Eq.10, is given as:

$$g(\alpha,\beta) = \alpha\tau - \beta\eta - (\alpha+\beta)\cdot\mathbb{E}_{\rho(s)}\left[-\frac{\alpha+\beta+\lambda}{\alpha+\beta}\right]. \tag{14}$$

As $\exp\left(\frac{\alpha+\beta+\lambda}{\alpha+\beta}\right)$ is the normalization term of $\pi(a|s)$, the dual function can be represented as:

$$g(\alpha,\beta) = \alpha\tau - \beta\eta + \mathbb{E}_{\rho(s)}\left[\alpha\cdot\log\pi_{\text{old}}(\boldsymbol{a}|\boldsymbol{s}) + Q(\boldsymbol{s},\boldsymbol{a}) - (\alpha+\beta)\cdot\log\pi(\boldsymbol{a}|\boldsymbol{s})\right]. \tag{15}$$

The approximate gradient of $g(\alpha,\beta)$ w.r.t. $\alpha$ and $\beta$ are:

$$\hat{\nabla}_\alpha g(\alpha) = \tau - \log\pi_\theta(\boldsymbol{a}|\boldsymbol{s}) + \log\pi_{\hat{\theta}}(\boldsymbol{a}|\boldsymbol{s}), \tag{16}$$

$$\hat{\nabla}_\beta g(\beta) = -\eta - \log\pi_\theta(\boldsymbol{a}|\boldsymbol{s}), \tag{17}$$

which enable us to find the "proper" $\alpha$ and $\beta$ with the gradient descent method, satisfying the KL and entropy constraints in Eq.2. The dual variable updates, along with the trust entropy Q function updates (Sec. 3.2) and trust entropy policy updates (Sec. 3.3), constitute the main components of our framework.

## 4 CONNECTION WITH PREVIOUS WORK

The most related methods to our work are MOTO (Akrour et al., 2016) , GAC (Tangkaratt et al., 2018), and Trust-PCL (Nachum et al., 2018) as they also consider both the trust region constraint and the entropy constraint.

MORE (Daniel et al., 2016) considers the two constraints in the domain of stochastic search optimization. MOTO (Akrour et al., 2016) extends MORE to the sequential decision making domain. In MOTO, Q function is estimated by using a quadratic surrogate function of the state and action space,

and the policy of a log-linear Gaussian form is updated according to the KL-divergence bounding constraint and a variable lower bound of entropy determined by the policy of each iteration.

GAC (Tangkaratt et al., 2018) further extends MOTO by 1) approximating the Q function with a truncated Taylor series expansion and parameterizing them with a deep neural network, and 2) learning log-nonlinear Gaussian form policies. In some sense, MOTO, GAC, and ours can be viewed as solving the optimization problem with the same constraints as stated in Eq.2. After leveraging the Lagrangian of the optimization problem, the corresponding closed form solution for the policy updating is shown in Eq.10. However, Eq.10 also indicates that the new policy is derived by weighing the old policy and the exponential of Q function (see the R.H.S of Eq.10), which may deviate the updated policy from the expected policy distribution class. Consequently, the KL constraint will be no longer preserved (Akrour et al., 2018). These methods differ from each other by using different strategies to circumvent the issue. MOTO utilizes a quadratic Q function and assumes the policy is of log-linear Gaussian form. Consequently, MOTO can update the policy in a non-parameterized way. GAC adopts the similar strategy as MOTO to learn a non-parameterized Gaussian actor, and then uses this actor to guide a parameterized actor with supervised learning. However, it is hard for MOTO and GAC to deal with high-dimensional action space because they rely on second-order computation. In comparison, we redefine the Bellman equation and guarantee the policy improvement by updating policies with Eq.11. Therefore, our method resolves the challenge simply with a more general assumption of Gaussian policy class.

Moreover, when dealing with the dual function

$$g(\alpha, \beta) = \alpha\tau - \beta\eta + (\alpha + \beta)\mathbb{E}_{\rho(s)}\left[\log\int\pi_{\text{old}}(\boldsymbol{a}|\boldsymbol{s})^{\frac{\alpha}{\alpha+\beta}}\exp\left(\frac{\hat{Q}(\boldsymbol{s}, \boldsymbol{a})}{\alpha + \beta}\right)\mathrm{d}\boldsymbol{a}\right], \qquad (18)$$

where the integral term is intractable, MOTO and GAC reply on complex second-order computation to make it tractable. In contrast, we resolve the challenge by leveraging the policy to transform the dual function into a simpler form which can still be optimized in first-order computation, e.g., stochastic gradient descent method, with the policy improvement guaranteed.

Trust-PCL (Nachum et al., 2018) addresses the challenge with a different perspective by integrating path consistency learning (PCL) (Nachum et al., 2017), which is developed in the maximum entropy framework, and trust region policy optimization method. PCL, which is the base algorithm of trust-PCL, suggests that the optimal policy and state values should satisfy pathwise soft consistency property along any sampled trajectory, thus allowing the use of off-policy data. Consequently, the single-step temporal consistency of state-value function in Trust-PCL is

$$V^*(s_t) = \mathbb{E}_{r_t, s_{t+1}}\left[r_t - (\tau + \lambda)\log\pi^*(a_t \mid s_t) + \lambda\log\tilde{\pi}(a_{t+i} \mid s_{t+i}) + \gamma V^*(s_{t+1})\right], \qquad (19)$$

which is similar to our state-value function. However, our method differs from Trust-PCL in 3 ways: 1) Trust-PCL focuses on updating the state-value function while our method focuses on updating the Q function. 2) Trust-PCL updates the policy directly with the temporal consistency squared error while we use Eq.11. 3) As each update iteration in Trust-PCL requires trajectory/sub-trajectory samples to satisfy the pathwise soft consistency, it significantly compromises the algorithm efficiency. In comparison, our method requires only state-action pairs for each update iteration and is capable of finding (sub-)optimal value for every dual variable in each update iteration.

It should be noted that proximal policy optimization (PPO) (Schulman et al., 2017) can also achieve trust region constraint and encourage unpredictably actions by adding entropy bonus to its loss function. However, the entropy bonus in PPO is one-off effect which only considers the current state but no future states of the agent. In comparison, our method can be considered as maximum long-term entropy with constraint policy in trust region.

## 5 EXPERIMENTS

We experimented to investigate the following questions: (1) In comparison with state-of-the-art algorithms, does TEAC have a better performance in terms of sample efficiency and computational efficiency? (2) How should we choose hyperparameters $\tau$ and $\eta$ and how can these two variables affect the performance?

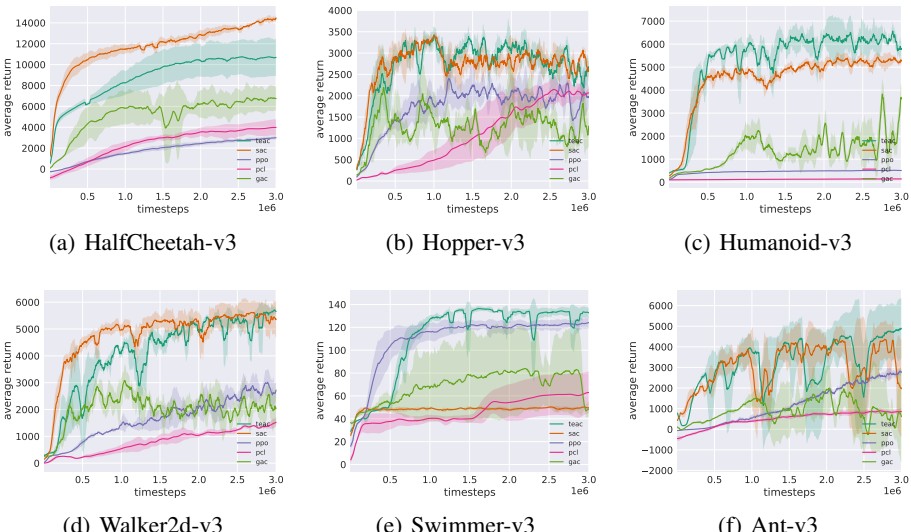

Figure 1: Performance comparisons on six MuJoCo tasks trained for 3 million timesteps. The horizontal axis indicates number of environment steps. The vertical axis indicates the average return. We trained three different instances of each algorithm with different random seeds, with each instance performing an evaluation every 4,000 environment steps. The solid lines represent the mean and the shaded regions mark the minimum and maximum returns over the three trials. We set $\eta$ as the negative of action space dimension of the task, and set $\tau = 0.005$ for all tasks.

## 5.1 SETUP

We experimented the continuous control tasks (HalfCheetah-v3, Hopper-v3, Humanoid-v3, Walker2d-v3, Swimmer-v3, Ant-v3) available from the MuJoCo environment (Todorov et al., 2012). We compared our method TEAC with 1) proximal policy optimization (PPO) (Schulman et al., 2017), a stable and effective on-policy policy gradient algorithm; 2) SAC (Haarnoja et al., 2018c), the state-of-the-art off-policy algorithm for learning maximum entropy policies whose temperature is adjusted automatically; 3) Trust-PCL (Nachum et al., 2018), an off-policy method optimizing maximum entropy RL objective with trust region; and 4) GAC (Tangkaratt et al., 2018), an off-policy method utilizing second-order information of critic. For Trust-PCL and GAC, we used their original implementation provided by their authors[3] [4] [5]. For PPO and SAC, we used the implementation publicly provided by OpenAI [6], and adapted SAC to its automatically adjusting version in Haarnoja et al. (2018c). For convenience, we developed our algorithm based on spinningup version of SAC[7]. The pseudo-code of our method is provided in Appendix C and the source code is available at https://github.com/ICLR2021papersub/TEAC.

TEAC requires to specify hyperparameters $\tau$, which represents the desired maximum KL-divergence, and $\eta$, which specifies the desired minimum entropy, before training. As the requirements of stability and exploration vary in different tasks, we set $\eta$ as the negative of action space dimension of the task, and set $\tau = 0.005$ for all tasks. The settings of effective hyperparameters are provided in Appendix D.

---

[3]Trust-PCL code: https://github.com/tensorflow/models/ tree/master/research/pcl_rl

[4]GAC code: https://github.com/voot-t/guide-actor-critic

[5]Due to the second-order computation complexity, we only finished testing GAC on HalfCheetah-v3 and Hopper-v3 at the time of paper submission. The experimental results show that we can achieve better performance than GAC in much shorter running time. We will complete the comparison before the rebuttal begins.

[6]OpenAI spinningup code: https://github.com/openai/spinningup

[7]https://github.com/openai/spinningup/tree/master/spinup/algos/pytorch/sac

## 5.2 RESULTS

Fig. 1 illustrates the training curve for each algorithm. In general, when $\tau = 0.005$, TEAC has similar performance as SAC in simpler tasks which have lower dimension of actions, such as Hopper-v3, Walker2d-v3, and Ant-v3. However, for complex tasks with higher dimension of actions, such as Huamoid-v3, TEAC gains significant performance improvement.

We also experimented and compared different settings of $\tau$ and $\eta$ for their impact on the performance. As there are no reasonable approaches to set $\tau$, we simply compared seven different value levels in the tasks. The results show that the selection of proper $\tau$ value for each task can significantly boost the performance of TEAC (see more details in Appendix B), and generally $\tau$ should be smaller for tasks with higher complexity. For example, when we set $\tau = 0.001$ in Humanoid-v3, TEAC had more than 10% performance gain than that of $\tau = 0.005$. The reason can be attributed to the stability of the algorithm. For complex tasks, the policy needs more exploration to find the global optima, resulting in larger update steps. Without the help of trust region constraint, the policy will explore arbitrarily in the policy space, losing its bearings and getting trapped in some bad settle-points.

For $\eta$, as we are dealing with continuous distributions, the entropy can be negative (Abdolmaleki et al., 2016). Hence, $\eta$ should be a small value. We have investigated several heuristic approaches for setting $\eta$ provided in MORE, GAC, and SAC, but none of them can serve as a general and effective solution (see more details in Appendix B). Thus, in our experiments, we simply set $\eta$ as the negative of action space dimension similar to SAC.

## 6 CONCLUSION

In this paper, we propose to integrate two branches of research in RL, trust region methods for better stability and maximum entropy methods for better policy exploration during the learning, to benefit from both sides. We first transform the original RL objective to a constraint optimization problem with the constraints of upper bound KL-divergence to avoid the abrupt difference between the new and old policies and lower bound entropy to promote the policy exploration. Therefore, the Bellman equation is redefined accordingly to guide the system loss evaluation. Consequently, we introduce TEAC, an off-policy algorithm to learn stable and sufficiently explored policies for continuous states and actions. TEAC utilizes two Actor networks to achieve the policy improvement by leveraging the information from the old policy and the exponential of current Q function represented in the critic network. The results show that TEAC with optimized parameters achieves similar performance in half of the tasks and notably improvement in the others in terms of efficiency and effectiveness.

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

# APPENDIX

## A DERIVATIONS AND PROOFS

### A.1 TRUST ENTROPY POLICY EVALUATION

**Lemma A.1** *(Trust Entropy Policy Evaluation). Let $Q^{k+1} = \mathcal{T}^\pi Q^k$, the sequence $Q^k$ will converge to the trust entropy Q-value of $\pi$ as $k \to \infty$ when considering a mapping $Q^0 : \mathcal{S} \times \mathcal{A} \to \mathbb{R}$ with $|\mathcal{A}| < \infty$ and the Bellman backup operator $\mathcal{T}^\pi$.*

*Proof.* We define the reward function in trust entropy framework as

$$r_\pi(\boldsymbol{s}, \boldsymbol{a}) \triangleq r(\boldsymbol{s}, \boldsymbol{a}) + (\alpha + \beta) \cdot \mathbb{E}_{\boldsymbol{s}' \sim p}\left[\mathbb{E}_{\alpha' \sim \pi}\left[\pi(\cdot|\boldsymbol{s}')\right]\right] - \alpha \cdot \mathbb{E}_{\boldsymbol{s}' \sim p}\left[\mathbb{E}_{\alpha' \sim \pi}\left[\pi_{\text{old}}(\cdot|\boldsymbol{s}')\right]\right]. \tag{20}$$

We rewrite the update rule as

$$Q(\boldsymbol{s}, \boldsymbol{a}) \leftarrow r_\pi(\boldsymbol{s}, \boldsymbol{a}) + \gamma \mathbb{E}_{\boldsymbol{s}' \sim p, \boldsymbol{a}' \sim \pi}\left[Q(\boldsymbol{s}', \boldsymbol{a}')\right]. \tag{21}$$

Following Sutton & Barto (1998), we can realize the standard convergence for policy evaluation.

### A.2 DERIVATION OF THE SOLUTION OF LAGRANGIAN

By taking derivative of $\mathcal{L}$ w.r.t. $\pi$ and setting the derivative to zero,

$$\begin{aligned}
\partial_\pi \mathcal{L} =& \mathbb{E}_{\rho(\boldsymbol{s})}\Bigg[\int \bigg(\hat{Q}(\boldsymbol{s}, \boldsymbol{a}) - (\alpha + \beta) \log \pi(\boldsymbol{a}|\boldsymbol{s}) + \alpha \log \pi_{\text{old}}(\boldsymbol{a}|\boldsymbol{s}) - \nu \hat{V}(\boldsymbol{s}) + \\
& \mathbb{E}_{p(\boldsymbol{s}'|\boldsymbol{s}, \boldsymbol{a})}[\nu \hat{V}(\boldsymbol{s}')]\bigg)\mathrm{d}\boldsymbol{a}\Bigg] - (\alpha + \beta + \lambda) \\
=& \hat{Q}(\boldsymbol{s}, \boldsymbol{a}) - (\alpha + \beta) \log \pi(\boldsymbol{a}|\boldsymbol{s}) + \alpha \log \pi_{\text{old}}(\boldsymbol{a}|\boldsymbol{s}) - \nu \hat{V}(\boldsymbol{s}) + \mathbb{E}_{p(\boldsymbol{s}'|\boldsymbol{s}, \boldsymbol{a})}[\nu \hat{V}(\boldsymbol{s}')] \\
& - (\alpha + \beta + \lambda) \\
=& 0
\end{aligned} \tag{22}$$

Given the fact that we sample actions from the actor network as parameterized Gaussian distribution and the value function should satisfy Bellman equation, if we set $\nu = 0$, the function can be rewritten as

$$0 = \hat{Q}(\boldsymbol{s}, \boldsymbol{a}) - (\alpha + \beta) \log \pi(\boldsymbol{a}|\boldsymbol{s}) + \alpha \log \pi_{\text{old}}(\boldsymbol{a}|\boldsymbol{s}) - (\alpha + \beta + \lambda) \tag{23}$$

The solution of $\pi(\boldsymbol{a}|\boldsymbol{s})$ is:

$$\pi(\boldsymbol{a}|\boldsymbol{s}) = \pi_{\text{old}}(\boldsymbol{a}|\boldsymbol{s})^{\frac{\alpha}{\alpha+\beta}} \cdot \exp\left(\frac{\hat{Q}(\boldsymbol{s}, \boldsymbol{a})}{\alpha + \beta}\right) \cdot \exp\left(-\frac{\alpha + \beta + \lambda}{\alpha + \beta}\right). \tag{24}$$

Here, we combine the third constraint in Eq. 2 with the solution Eq. 24:

$$\begin{aligned}
1 =& \mathbb{E}_{\rho(\boldsymbol{s})}\left[\int \pi(\boldsymbol{a}|\boldsymbol{s}) da\right] \\
=& \mathbb{E}_{\rho(\boldsymbol{s})}\left[\int \pi_{\text{old}}(\boldsymbol{a} \mid \boldsymbol{s})^{\frac{\alpha}{\alpha+\beta}} \cdot \exp\left(\frac{\hat{Q}(\boldsymbol{s}, \boldsymbol{a})}{\alpha + \beta}\right) \cdot \exp\left(-\frac{\alpha + \beta + \lambda}{\alpha + \beta}\right) da\right].
\end{aligned} \tag{25}$$

Given the fact that $\alpha$, $\beta$, and $\lambda$ are constants which are independent of $\boldsymbol{s}$ and $\boldsymbol{a}$, we can get

$$\begin{aligned}
1 =& \mathbb{E}_{\rho(\boldsymbol{s})}\left[\int \pi_{\text{old}}(\boldsymbol{a} \mid \boldsymbol{s})^{\frac{\alpha}{\alpha+\beta}} \cdot \exp\left(\frac{\hat{Q}(\boldsymbol{s}, \boldsymbol{a})}{\alpha + \beta}\right) \cdot \exp\left(-\frac{\alpha + \beta + \lambda}{\alpha + \beta}\right) da\right] \\
=& \mathbb{E}_{\rho(\boldsymbol{s})}\left[\int \pi_{\text{old}}(\boldsymbol{a} \mid \boldsymbol{s})^{\frac{\alpha}{\alpha+\beta}} \cdot \exp\left(\frac{\hat{Q}(\boldsymbol{s}, \boldsymbol{a})}{\alpha + \beta}\right) da\right] \cdot \exp\left(-\frac{\alpha + \beta + \lambda}{\alpha + \beta}\right),
\end{aligned} \tag{26}$$

Thus,

$$\exp\left(-\frac{\alpha+\beta+\lambda}{\alpha+\beta}\right)^{-1} = \mathbb{E}_{\rho(\boldsymbol{s})}\left[\int \pi_{\text{old}}\left(\boldsymbol{a}\mid\boldsymbol{s}\right)^{\frac{\alpha}{\alpha+\beta}} \cdot \exp\left(\frac{\hat{Q}(\boldsymbol{s},\boldsymbol{a})}{\alpha+\beta}\right) da\right]. \tag{27}$$

Hence, the term $\exp\left(\frac{\alpha+\beta+\lambda}{\alpha+\beta}\right)$ acts as a normalization term. Therefore,

$$\pi(\boldsymbol{a}|\boldsymbol{s}) \propto \pi_{\text{old}}(\boldsymbol{a}|\boldsymbol{s})^{\frac{\alpha}{\alpha+\beta}} \exp\left(\frac{\hat{Q}(\boldsymbol{s},\boldsymbol{a})}{\alpha+\beta}\right). \tag{28}$$

## A.3 TRUST ENTROPY POLICY IMPROVEMENT

**Lemma A.2** *(Trust Entropy Policy Improvement). Given a policy $\pi$ and an old policy $\hat{\pi}$, define a new policy*

$$\tilde{\pi}(\boldsymbol{a}|\boldsymbol{s}) \propto \hat{\pi}(\boldsymbol{a}|\boldsymbol{s})^{\frac{\alpha}{\alpha+\beta}} \exp\left(\frac{Q^{\pi}(\boldsymbol{s},\boldsymbol{a})}{\alpha+\beta}\right), \quad \forall \boldsymbol{s}. \tag{29}$$

*If $Q$ is bounded, $\int \hat{\pi}(\boldsymbol{a}|\boldsymbol{s})^{\frac{\alpha}{\alpha+\beta}} \exp\left(\frac{Q^{\pi}(\boldsymbol{s},\boldsymbol{a})}{\alpha+\beta}\right) d\boldsymbol{a}$ is bounded for any $\boldsymbol{s}$ (for all $\pi$, $\hat{\pi}$ and $\tilde{\pi}$), and the policies are the parameterized Gaussian networks. Then we can obtain $Q^{\tilde{\pi}}(\boldsymbol{s},\boldsymbol{a}) \geq Q^{\pi}(\boldsymbol{s},\boldsymbol{a}), \forall \boldsymbol{s},\boldsymbol{a}$.*

*Proof.* With the definition of the $V$ function, we can get:

$$V^{\pi}(\boldsymbol{s}) = \mathbb{E}_{\tau\sim\pi,s_0=s,a_0=a}\left[\sum_{t=0}^{\infty} \gamma^t(r(s_t,a_t) - (\alpha+\beta)\cdot\log\pi(\cdot|s_t) + \alpha\cdot\log\hat{\pi}(\cdot|s_t))\right] \tag{30}$$

Here, $\tau = (s_0,a_0,s_1,a_1,\dots)$ denotes the trajectory originating at $(\boldsymbol{s},\boldsymbol{a})$. Using a telescoping argument, we have:

$$V^{\tilde{\pi}}(s) - V^{\pi}(s) = \mathbb{E}_{\tau\sim\tilde{\pi},s_0=s,a_0=a}\left[\sum_{t=0}^{\infty} \gamma^t(r(s_t,a_t) - (\alpha+\beta)\cdot\log\tilde{\pi}(\cdot|s_t) + \alpha\cdot\log\pi(\cdot|s_t))\right] - V^{\pi}(s)$$

$$= \mathbb{E}_{\tau\sim\tilde{\pi},s_0=s,a_0=a}[\sum_{t=0}^{\infty} \gamma^t(r(s_t,a_t) - (\alpha+\beta)\cdot\log\tilde{\pi}(\cdot|s_t) + \alpha\cdot\log\pi(\cdot|s_t)$$
$$+ V^{\pi}(s_t) - V^{\pi}(s_t))] - V^{\pi}(s)$$

$$\overset{(a)}{=} \mathbb{E}_{\tau\sim\tilde{\pi},s_0=s,a_0=a}[\sum_{t=0}^{\infty} \gamma^t(r(s_t,a_t) - (\alpha+\beta)\cdot\log\tilde{\pi}(\cdot|s_t) + \alpha\cdot\log\pi(\cdot|s_t)$$
$$+ \gamma V^{\pi}(s_{t+1}) - V^{\pi}(s_t))]$$

$$\overset{(b)}{=} \mathbb{E}_{\tau\sim\tilde{\pi},s_0=s,a_0=a}[\sum_{t=0}^{\infty} \gamma^t(r(s_t,a_t) - (\alpha+\beta)\cdot\log\tilde{\pi}(\cdot|s_t) + \alpha\cdot\log\pi(\cdot|s_t)$$
$$+ \gamma\mathbb{E}[V^{\pi}(s_{t+1})|s_t,a_t] - V^{\pi}(s_t))]$$

$$= \mathbb{E}_{\tau\sim\tilde{\pi},s_0=s,a_0=a}[\sum_{t=0}^{\infty} \gamma^t(Q^{\pi}(s_t,a_t) - V^{\pi}(s_t) - (\alpha+\beta)\cdot\log\tilde{\pi}(\cdot|s_t) + \alpha\cdot\log\pi(\cdot|s_t))]$$

$$= \frac{1}{1-\gamma}\mathbb{E}_{s'\sim\rho_{\tilde{\pi}}(s)}\mathbb{E}_{a\sim\tilde{\pi}(\cdot|s)}[\gamma^t(Q^{\pi}(s',a) - V^{\pi}(s') - (\alpha+\beta)\cdot\log\tilde{\pi}(\cdot|s') + \alpha\cdot\log\pi(\cdot|s'))],$$
$$\tag{31}$$

where (a) rearranges terms in the summation and cancels the $V(s_0)$ term with the $-V(s)$ outside the summation, and (b) uses the tower property of conditional expectations and the final equality follows from the definition of $\rho_{\tilde{\pi}}(s)$. Consider Eq.24 can be rewritten as:

$$\pi(\boldsymbol{a}|\boldsymbol{s}) = \exp\left(\frac{\alpha}{\alpha+\beta}\log\hat{\pi}(\boldsymbol{a}|\boldsymbol{s}) + \frac{Q^{\pi}(\boldsymbol{s},\boldsymbol{a})}{\alpha+\beta}\right)\exp\left(-\frac{\alpha+\beta+\lambda}{\alpha+\beta}\right), \tag{32}$$

where $\exp\left(-\frac{\alpha+\beta+\lambda}{\alpha+\beta}\right)$ is normalization term. Assume we follow the gradient ascent update rule and that the distribution $\rho(s)$ is strictly positive i.e. $\rho(s) > 0$ for all states s. Following the work Agarwal et al. (2020), with the help of the gradient of the softmax policy class, we can get

$$\sum_{a \in \mathcal{A}} \tilde{\pi}(a|s)(Q(s,a) - V(s) - (\alpha + \beta) \cdot \log \tilde{\pi}(\cdot|s) + \alpha \cdot \log \pi(\cdot|s)) \geq 0, \tag{33}$$

Then we get $V^{\tilde{\pi}}(s) \geq V^{\pi}(s)$, as well as $Q^{\tilde{\pi}}(s,a) \geq Q^{\pi}(s,a)$ holds for all states $s$ and actions $a$.

### A.4 DERIVATION OF THE DUAL FUNCTION

To obtain the dual function, we take the solution of $\pi(a|s)$ into Eq. 3:

$$
\begin{aligned}
\mathcal{L}(\pi, \alpha, \beta, \lambda) =& \mathbb{E}_{\rho(\boldsymbol{s})\pi(\boldsymbol{a}|\boldsymbol{s})}[\hat{Q}(\boldsymbol{s}, \boldsymbol{a})] + \alpha \left(\tau - \mathbb{E}_{\rho(\boldsymbol{s})}\left[\mathrm{KL}\left(\pi(\cdot|\boldsymbol{s})\|\pi_{\mathrm{old}}(\cdot|\boldsymbol{s})\right)\right]\right) \\
&+ \beta \left(\mathbb{E}_{\rho(\boldsymbol{s})}[\mathrm{H}(\pi(\cdot|\boldsymbol{s}))] - \eta\right) + \lambda \left(\mathbb{E}_{\rho(\boldsymbol{s})} \int \pi(\boldsymbol{a}|\boldsymbol{s}) d\boldsymbol{a} - 1\right) \\
=& \mathbb{E}_{\rho(\boldsymbol{s})\pi(\boldsymbol{a}|\boldsymbol{s})}[\hat{Q}(\boldsymbol{s}, \boldsymbol{a})] \\
&- (\alpha + \beta)\mathbb{E}_{\rho(\boldsymbol{s})\pi(\boldsymbol{a}|\boldsymbol{s})}\left[\frac{\hat{Q}(\boldsymbol{s}, \boldsymbol{a})}{\alpha + \beta} + \frac{\alpha}{\alpha + \beta}\log \pi_{\mathrm{old}}(\boldsymbol{a}|\boldsymbol{s}) - \frac{\alpha + \beta + \lambda}{\alpha + \beta}\right] \\
&+ \alpha\mathbb{E}_{\rho(\boldsymbol{s})\pi(\boldsymbol{a}|\boldsymbol{s})}\left[\log \pi_{\mathrm{old}}(\boldsymbol{a}|\boldsymbol{s})\right] + \lambda \left(\mathbb{E}_{\rho(\boldsymbol{s})} \int \pi(\boldsymbol{a}|\boldsymbol{s}) d\boldsymbol{a} - 1\right) + \alpha\tau - \beta\eta \\
=& \alpha\tau - \beta\eta - (\alpha + \beta) \cdot \mathbb{E}_{\rho(\boldsymbol{s})}\left[-\frac{\alpha + \beta + \lambda}{\alpha + \beta}\right].
\end{aligned}
\tag{34}
$$

This loss function can be rewritten as

$$
\begin{aligned}
\mathcal{L}(\alpha, \beta) =& \alpha\tau - \beta\eta + (\alpha + \beta)\mathbb{E}_{\rho(s)}\left[\log\left(\exp\left(\frac{\alpha + \beta + \lambda}{\alpha + \beta}\right)\right)\right] \\
=& \alpha\tau - \beta\eta + (\alpha + \beta)\mathbb{E}_{\rho(s)}\left[\log \int \pi_{\mathrm{old}}(\boldsymbol{a}|\boldsymbol{s})^{\frac{\alpha}{\alpha+\beta}} \exp\left(\frac{\hat{Q}(\boldsymbol{s}, \boldsymbol{a})}{\alpha + \beta}\right) d\boldsymbol{a}\right] \\
=& g(\alpha, \beta)
\end{aligned}
\tag{35}
$$

Meanwhile, we can rewrite Eq.24 as

$$
\exp\left(\frac{\alpha + \beta + \lambda}{\alpha + \beta}\right) = \frac{\pi_{\mathrm{old}}(\boldsymbol{a}|\boldsymbol{s})^{\frac{\alpha}{\alpha+\beta}} \cdot \exp\left(\frac{\hat{Q}(\boldsymbol{s},\boldsymbol{a})}{\alpha+\beta}\right)}{\pi(\boldsymbol{a}|\boldsymbol{s})}
\tag{36}
$$

With Eq.36, the loss function becomes

$$
\begin{aligned}
\mathcal{L}(\alpha, \beta) =& \alpha\tau - \beta\eta + (\alpha + \beta)\mathbb{E}_{\rho(s)}\left[\log\left(\frac{\pi_{\mathrm{old}}(\boldsymbol{a}|\boldsymbol{s})^{\frac{\alpha}{\alpha+\beta}} \cdot \exp\left(\frac{\hat{Q}(\boldsymbol{s},\boldsymbol{a})}{\alpha+\beta}\right)}{\pi(\boldsymbol{a}|\boldsymbol{s})}\right)\right] \\
=& \alpha\tau - \beta\eta + (\alpha + \beta)\mathbb{E}_{\rho(s)}\left[\frac{\alpha}{\alpha + \beta}\log \pi_{\mathrm{old}}(\boldsymbol{a}|\boldsymbol{s}) + \frac{\hat{Q}(\boldsymbol{s},\boldsymbol{a})}{\alpha+\beta} - \log \pi(\boldsymbol{a}|\boldsymbol{s})\right] \\
=& \alpha\tau - \beta\eta + \mathbb{E}_{\rho(s)}\left[\alpha \cdot \log \pi_{\mathrm{old}}(\boldsymbol{a}|\boldsymbol{s}) + \hat{Q}(\boldsymbol{s},\boldsymbol{a}) - (\alpha + \beta) \cdot \log \pi(\boldsymbol{a}|\boldsymbol{s})\right].
\end{aligned}
\tag{37}
$$

# B HYPERPARAMETER ANALYSIS FOR $\tau$ AND $\eta$

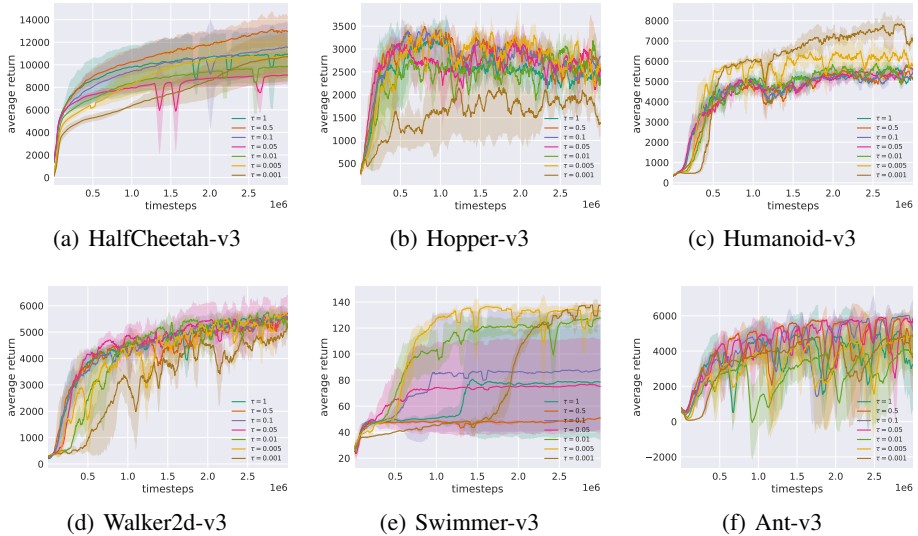

(a) HalfCheetah-v3      (b) Hopper-v3      (c) Humanoid-v3

(d) Walker2d-v3      (e) Swimmer-v3      (f) Ant-v3

Figure 2: Performance with different $\tau$ on six MuJoCo tasks. $\tau = 0.001$ achieves about 8000 as the return in Humanoid-v3, and $\tau = 0.005$ achieves about 130 as the return in Swimmer-v3. These two results surpass all benchmark algorithms significantly.

The impact of $\tau$ on the performance of TEAC was evaluated with $\tau \in (0.001, 0.005, 0.01, 0.05, 0.1, 0.5, 1)$. Fig.5 shows that tuning $\tau$ for different tasks may achieve significant performance improvement in TEAC.

Similar to SAC (Haarnoja et al., 2018c), we set $\eta$ to the negative of action space dimension in our own experiments. Besides, there are other techniques in existing methods. MORE (Abdolmaleki et al., 2016) changes the entropy constraint to

$$E - E_0 \geq \gamma \left(E_{\text{old}} - E_0\right) \Rightarrow \eta = \gamma \left(E_{\text{old}} - E_0\right) + E_0 \tag{38}$$

where $E \approx \mathbb{E}_{\rho(\boldsymbol{s})} \left[\text{H}\left(\pi_{\boldsymbol{\theta}}(\boldsymbol{a}|\boldsymbol{s})\right)\right]$ denotes the expected entropy of the current policy, $E_{\text{old}} \approx \mathbb{E}_{\rho(\boldsymbol{s})} \left[\text{H}\left(\pi_{\hat{\boldsymbol{\theta}}}(\boldsymbol{a}|\boldsymbol{s})\right)\right]$ denotes the expected entropy of the old policy, and $E_0$ denotes the entropy of a base policy $\mathcal{N}(\boldsymbol{a}|\boldsymbol{0}, 0.01\mathbf{I})$. GAC improves this technique to adjust $\eta$ heuristically by

$$\eta = \max\left(\gamma\left(E - E_0\right) + E_0, E_0\right). \tag{39}$$

We compared these three different techniques on Hopper-v3, Humanoid-v3 and Ant-v3. Fig. 3 shows that there is no outstanding difference from them. Therefore, we simply set $\eta$ as the negative of action space dimension similar to SAC.

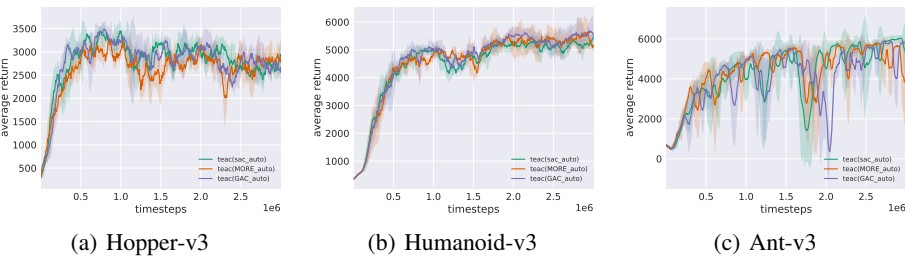

(a) Hopper-v3      (b) Humanoid-v3      (c) Ant-v3

Figure 3: Performance with different $\eta$ in three Mujoco tasks with $\tau = 0.1$.

## C  Algorithms

---

**Algorithm 1** TEAC: Trust Entropy Actor Critic

---

**Input:**
   Initial actor $\pi_\theta(a|s)$, old actor $\pi_{\hat{\theta}}(a|s)$,
   critic $Q_{\phi_1}$ and $Q_{\phi_2}$[8], target critic network $\bar{\phi}_1 \leftarrow \phi_1$, $\bar{\phi}_2 \leftarrow \phi_2$,
   KL divergence bound $\tau$, entropy bound $\eta$, learning rate $\omega_{\text{ac}}$, $\omega_\alpha$, $\omega_\beta$,
   an empty replay pool $\mathcal{D} = \varnothing$

1: **for** each iteration **do**
2:    **for** each environment step **do**
3:        Observe state $s_t$ and sample action $a_t \sim \pi_\theta(a_t|s_t)$
4:        Execute $a_t$, receive reward $r(s_t, a_t)$ and next state $s'_t \sim p(s'_t|s_t, a_t)$
5:        Add transition to replay buffer $\mathcal{D} \leftarrow \mathcal{D} \cup \{(\mathbf{s}_t, \mathbf{a}_t, r(\mathbf{s}_t, \mathbf{a}_t), s'_t)\}$
6:    **end for**
7:    **for** each gradient step **do**
8:        Sample $N$ mini-batch samples $\{(s_i, a_i, r_i, s'_i)\}_{i=1}^N$ uniformly from $\mathcal{D}$
9:        Sample actions $a'_i \sim \pi_\theta(a|s'_i)$, compute $Q_{\text{tar}}(s'_i, a'_i)$:

$$Q_{\text{tar}}(s'_i, a'_i) = \min(Q_{\bar{\phi}_1}(s'_i, a'_i), Q_{\bar{\phi}_2}(s'_i, a'_i)) \tag{40}$$

10:        Compute $y_i$, update $\phi$ by, e.g., Adam, and update $\bar{\phi}$ by moving average:

$$y_i = r_i + \gamma \left(Q_{\text{tar}}(s'_i, a'_i) - (\alpha + \beta)\log \pi_\theta(a|s'_i) + \alpha \log \pi_{\hat{\theta}}(a|s'_i)\right) \tag{41}$$

$$\phi_j \leftarrow \phi_j - \omega_{\text{ac}} \nabla_{\phi_j} \frac{1}{N} \sum_{i=1}^N \left(Q_{\phi_j}(s_i, a_i) - y_i\right)^2, j \in \{1, 2\} \tag{42}$$

$$\bar{\phi}_j \leftarrow \kappa \phi_j + (1 - \kappa)\bar{\phi}_j, j \in \{1, 2\} \tag{43}$$

11:        Sample actions $\tilde{a} \sim \pi_\theta(a|s_i)$, compute $Q(s_i, \tilde{a})$:

$$Q(s_i, \tilde{a}) = \min(Q_{\phi_1}(s_i, \tilde{a}), Q_{\phi_2}(s_i, \tilde{a})) \tag{44}$$

12:        Compute loss function of $\theta$:

$$\mathcal{L}(\theta) = \frac{1}{N} \sum_{i=1}^N \left((\alpha + \beta)\log \pi_\theta(a|s_i) - \alpha \log \pi_{\hat{\theta}}(a|s_i) - Q(s_i, \tilde{a})\right) \tag{45}$$

13:        Update $\hat{\theta}$ using $\hat{\theta} \leftarrow \theta$
14:        Update $\theta$ by, e.g., Adam:
$$\theta \leftarrow \theta - \omega_{\text{ac}} \nabla_\theta \mathcal{L}(\theta) \tag{46}$$
15:        Compute loss function of dual variables $\alpha$ and $\beta$:

$$\mathcal{L}(\alpha) = -\frac{1}{N} \sum_{i=1}^N \alpha \cdot \left(\log \pi_\theta(a|s_i) - \log \pi_{\hat{\theta}}(a|s_i) - \tau\right) \tag{47}$$

$$\mathcal{L}(\beta) = -\frac{1}{N} \sum_{i=1}^N \beta \cdot \left(\log \pi_\theta(a|s_i) + \eta\right) \tag{48}$$

16:        Update dual variables $\alpha$ and $\beta$:
$$\alpha \leftarrow \alpha - \omega_\alpha \nabla_\alpha \mathcal{L}(\alpha), \beta \leftarrow \beta - \omega_\beta \nabla_\beta \mathcal{L}(\beta) \tag{49}$$

17:    **end for**
18: **end for**

---

# D HYPERPARAMETERS

Table 1 lists the effective hyperparameters of TEAC used in the experiments, of which the results are shown in Fig. 1.

Table 1: TEAC Hyperparameters

| Parameter | Value |
|---|---|
| optimizer | Adam (Kingma & Ba, 2015) |
| learning rate for actor and critic | $1 \cdot 10^{-3}$ |
| learning rate for $\alpha$ | $1 \cdot 10^{-4}$ |
| learning rate for $\beta$ | $1 \cdot 10^{-3}$ |
| discount ($\gamma$) | 0.99 |
| replay buffer size | $10^6$ |
| number of hidden layers (all networks) | 2 |
| number of hidden units per layer | 256 |
| number of samples per minibatch | 100 |
| target entropy ($\eta$) | -dim($\mathcal{A}$))(e.g., -6 for HalfCheetah-v3) |
| max divergence for KL ($\tau$) | 0.005 |
| nonlinearity | ReLU |
| target smoothing coefficient $\kappa$ | 0.005 |
| target update interval | 1 |
| gradient steps | 1 |

# E THE BEST PERFORMANCE OF OUR MODEL

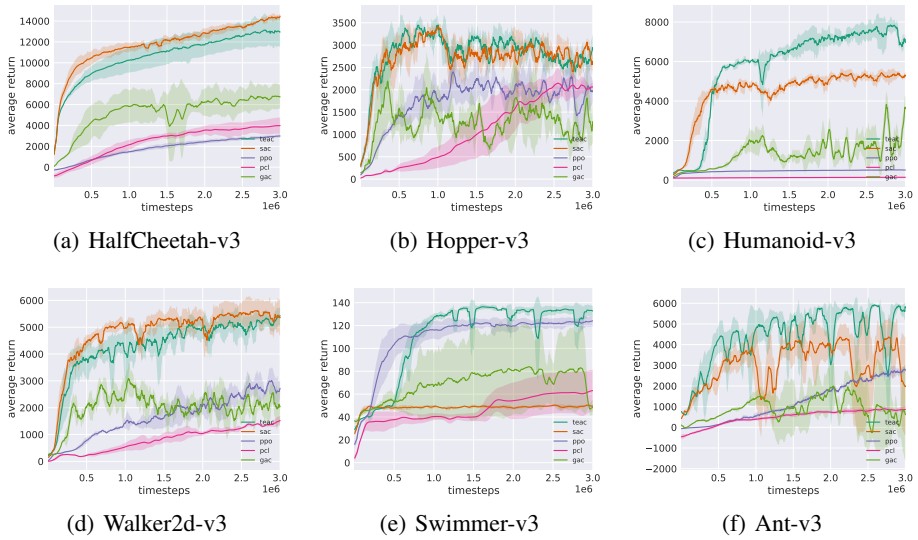

(a) HalfCheetah-v3     (b) Hopper-v3     (c) Humanoid-v3

(d) Walker2d-v3     (e) Swimmer-v3     (f) Ant-v3

Figure 4: Performance comparisons on six MuJoCo tasks. Notice that the blue line is the performance of our model which setting different $\tau$ with respect to different tasks. In this figure, we set $\tau = 0.5$ for HalfCheetah-v3, $\tau = 0.05$ for Ant-v3, $\tau = 0.1$ for Hopper-v3, $\tau = 0.001$ for Humanoid-v3, $\tau = 0.005$ for Swimmer-v3, and $\tau = 0.1$ for Walker2d-v3.

---

[8]Our implementation also makes use of two Q-functions (critic networks) to mitigate positive bias in the policy improvement step , follows Fujimoto et al. (2018) and Haarnoja et al. (2018b) .

# F ADDITIONAL CONTINUOUS CONTROL EXPERIMENTS WITH SIX INSTANCES

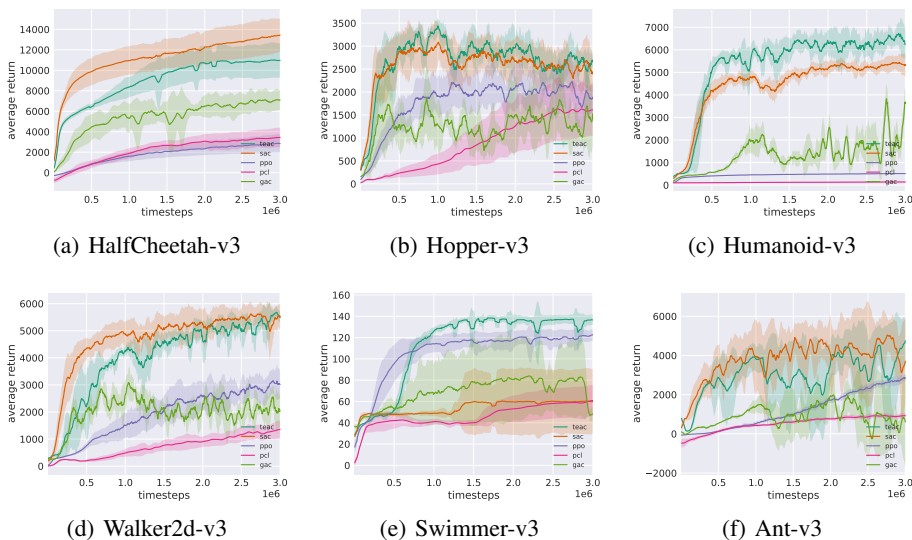

(a) HalfCheetah-v3     (b) Hopper-v3     (c) Humanoid-v3

(d) Walker2d-v3     (e) Swimmer-v3     (f) Ant-v3

Figure 5: Performance comparisons on six MuJoCo tasks. We trained six different instances of all algorithms with different random seeds. In this case, for TEAC, we set $\eta$ as the negative of action space dimension of the task, and set $\tau = 0.005$ for all tasks.

