# OpenReview forum: "TEAC: Intergrating Trust Region and Max Entropy Actor Critic for Continuous Control"
_ICLR.cc/2021/Conference — Reject_

### Official Review · AnonReviewer3 · 2020-10-28
**Possibly a good paper**

**Rating:** 7
**Confidence:** 1

**Review:**

The paper addresses the problem of reinforcement learning in continuous spaces by formulating the problem as a constrained optimization problem. In this problem, the objective is maximizing the expected reward, and the constraints ensure that 1) the distance between the new and old policies is bounded, 2) the entropy is above a threshold, and 3) the assumptions of MDP hold. The paper then derives closed-form solutions for the Lagrangian, which are used for obtaining variable update rules. An empirical study examines the ideas against benchmark problems.

The paper has a good structure and style. The claims seem sound and the results look promising. However, since I'm not familiar with the topic, I cannot verify the details of those claims and results. I base my recommendation on a high-level idea of the paper.

---

> ### Author Response · Authors · 2020-11-15
> **Response to Reviewer #3**
>
> Thank you very much for the recognition!

---

### Official Review · AnonReviewer2 · 2020-10-28
**Hard to follow work proposing a trust region policy optimization model with additional constraints**

**Rating:** 5
**Confidence:** 2

**Review:**

The authors propose to add three more constraints to the well-known trust region policy optimization model. The first one of these constraints aims at keeping the entropy level higher than a given threshold. The authors then relax these constraints to obtain the Lagrangian function. An approach using primal-dual updates is proposed to obtain a solution.

In general, I find the paper difficult to follow and hence cannot properly assess its novelty. My comments and questions are as follows:

- The function Q^\pi in (1) is not defined at that stage.

- How can we guarantee the feasibility of model (2) for different values of \eta? Likewise, if we obtain a solution with tackling the Lagrangian function, then is that solution feasible for (2)?

- How do you go from the first line to the second line in relation (4)?

- Why \lambda is in (5)?

- What is Q-\phi in (8)?

- What does setting \eta to dimension in Table 1 signify?

---

> ### Author Response · Authors · 2020-11-15
> **Response to Reviewer #2**
>
> We thank the reviewer for the very detailed and constructive feedback! We have updated the paper and highlighted the corrections/detailed illustrations following the reviewer's comments.
>
> 1. We have updated the paper with formal definitions of $Q^\pi$ and $V^\pi$, as well as the corresponding explanations, in Section 2. The changes are highlighted in the revision.
> 2. $\eta$ is one of the hyperparameters used in TEAC. As a constraint, there is no guarantee that we can always find a valid solution for an arbitrary $\eta$. Therefore, we followed SAC[I] to use the negative of action space dimension as a heuristic setting of $\eta$ in our experiments, indicating to what extent we encourage the policy to explore. Such a setting is usually less than the entropy of a random policy drawn from a Gaussian distribution for most tasks as we observed in the experiments. Once we can find a solution tackling the Lagrangian function for a specific $\eta$, then the solution should be feasible for (2).
> 3. There are two ways to derive the second line of Eq.4. 1) Since $\alpha+\beta+\lambda$ is a constant, we can move the term into the expectation and integral. Because we can add bounded constant to the Q function without changing the solution of Eq.2, we assume that Q is shifted so that the term inside the integral is non-negative. In this case, the expectation equals zero if and only if the term inside is zero (Eq.4). 2) Consider the derivative is with respect to $\pi(a|s)$ for each pair $(s,a)$. In this case, given a specific pair of $(s, a)$, the derivative is zero for other states and actions. We will add the explanation to the paper.
> 4. We have removed $\lambda$ from Eq.5 in the updated paper.
> 5. $Q_{\phi}$ represents the Q network (also known as critic network) which is parameterized by $\phi$. We have updated the paper to clarify the symbols in the paper.
> 6. We followed SAC to use the negative of action space dimension as a heuristic setting of $\eta$ in our experiments, indicating to what extent we encourage the policy to explore.
> In Appendix, we did provide the comparison with different heuristics of $\eta$'s settings suggested in the literature, including 1) setting $\eta$ as the negative of action space dimension (SAC), 2) the decreasing version (Eq.39) as in MORE[II], and 3) a GAC[III] setting (Eq.40). However, the difference regarding the performance is not noticeable (Fig.3 in Appendix B). It is probably because that the negative action space is usually less than $E_0$ which is the entropy of the randomized policy drawn from Gaussian distribution at the beginning and the starting entropy threshold in Eq.39 for most cases as we observed in the experiments. Therefore, we simply set $\eta$ as the negative of the action space dimension similar to SAC for our all experiments.
>
> [I]: Tuomas Haarnoja, Aurick Zhou, Pieter Abbeel, Sergey Levine. Soft Actor-Critic: Off-Policy Maximum Entropy Deep Reinforcement Learning with a Stochastic Actor. ICML 2018: 1856-1865
>
> [II]: Abbas Abdolmaleki, Rudolf Lioutikov, Jan Peters, Nuno Lau, Luís Paulo Reis, Gerhard Neumann. Model-Based Relative Entropy Stochastic Search. NIPS 2015: 3537-3545
>
> [III]: Voot Tangkaratt, Abbas Abdolmaleki, Masashi Sugiyama. Guide Actor-Critic for Continuous Control. ICLR (Poster) 2018

---

### Official Review · AnonReviewer4 · 2020-10-31
**An interesting algorithm and and analysis, but aspects of the paper need improvement**

**Rating:** 5
**Confidence:** 3

**Review:**

Previous work such as MOTO and GAC has developed approaches that combine trust regions for policy stability with maximum entropy to ensure adequate exploration.  This paper introduces an algorithm called TEAC in this spirit which adds several technical novelties to address limitations of previous work:
 - It uses a particular Gaussian policy parameterization for the actor to ensure that updated policies are easily representable in the same class while prior approaches used more complex solutions.
 - It uses a modified objective for the critic that incorporates the trust region and entropy terms
 - It uses a first (rather than second) order method to optimize the dual variables
 - It uses these to provide a proof of policy improvement.
TEAC is evaluated on a number of MuJoCo tasks.

I think the TEAC algorithm and theoretical analysis are a nice contribution and that ultimately this will be a nice paper.  However, I have some concerns with the positioning of the contribution, the rigor of the technical exposition, and the experiments.

1) The introduction makes it seem like the contribution of the paper is proposing the idea of combining trust regions and maximum entropy, but as is discussed later this idea is already present in previous work (e.g MOTO and GAC).  The real contribution is discussed only in very compressed form in the last paragraph.  Having a fuller discussion in Section 4 after the technical exposition is reasonable, but the key ideas and problems the paper solves to make this approach work better than previous work should at least by clearly stated and some intuition or motivation provided in the introduction.

2) In Equation (2), something about the notation for the =1 constraint doesn’t seem quite right.  The expectation isn’t being taken over anything.  This issue persists into 3.  The notation for the last MDP constraint seems a bit odd as well, although I supposes the intent is to have s’ drawn from the resulting distribution.

3) Equation 5 still has \lambda in it despite setting it to 0.

4) Lemma 2: the old policy \hat{\pi} is being given, not defined right?  At least I don’t see a definition of it.

5) The proof of Lemma 2 should point to A.3 not A.2.  I’m not sure why A.2 exists.  It doesn’t seem to be otherwise referenced and seems to be a repetition of material in the main text.

6) The reason (27) is true should be explained rather than point generally to another paper without an specific explanation.  I also don’t understand why (28) is an application of (6) and there is an ellipsis here.  I assume what is going on is an argument about what happens as the Bellman operator is repeatedly applied as it approaches its limit, which we know from (10) is the final quantity.

7) Given the close relationship, it seems odd that no comparison to MOTO is made.  GAC does not appear to provide one either, so even an indirect comparison does not appear to exist.  Furthermore GAC mysteriously disappears after Figure 1(b), which is problematic as it appears to be the most closely related work in terms of technique.

8) The plots only show three trials of each algorithm, which limits the confidence of performance assessments based on them and is not what I would describe as “extensive” in the abstract and elsewhere.  While I agree based on the results shown that TEAC typically outperforms the non-SAC algorithms, the comparison with SAC is more mixed except for the strong performance in swimmer (although even here the strong performance of PPO seems similar to TEAC).  So I would tone down the unqualified claim that “TEAC outperforms the state-of-the-art solutions”.

Updates after author responses and discussion:
(1) After the updates, most of the positioning issues have been improved
(2) The technical exposition is improved, and the main argument I was bothered by is somewhat improved, though it still has a big jump near the end.
(3) I'm still bothered that to demonstrate improvement the algorithm is tuned on a per-example basis but the baselines are not.  The results certainly show that the former is reasonable, but to make the comparison fair the latter needs to be done as well.

---

> ### Author Response · Authors · 2020-11-15
> **Response to Reviewer #4**
>
> We thank the reviewer for the very detailed and constructive feedback! We have updated the paper and highlighted the corrections/detailed illustrations following the reviewer's comments.
>
> 1. We are reorganizing the paper and detailing the discussion as the reviewer suggested. The new version will be uploaded soon.
> 2. We have updated the descriptions regarding these two constraints in the paper. Besides, to clarify the expected reward, we updated Eq.1 and Eq.2 with $\rho_{\pi}(s,a) = \rho_{\pi}(s)\pi(a|s)$, representing the joint distribution. The purpose of the third constraint is to ensure that the distribution is a proper probability density function. Regarding the fourth constraint, as the state marginal of the trajectory distribution needs to comply with the policy $\pi(a|s)$ and the system dynamics $p(s'|s, a)$, i.e., $\rho_{\pi}(s) \pi(a | s) p(s'|s, a) = \rho_{\pi}(s')$, meanwhile the direct matching of the state probabilities is not feasible in continuous state spaces, the use of $V^{\pi}(s')$, which can be also considered as state features, helps to focus on matching the feature averages.
> 3. We have removed $\lambda$ in Eq.5.
> 4. An old policy in the TEAC framework refers to a policy from the last iteration, which is, in practice, parameterized by the old policy network $\pi_{\hat{\theta}}$. And the old policy network $\pi_{\hat{\theta}}$ is copied from the policy network $\pi_{\theta}$ of the last iteration. We will further clarify it in the paper.
> 5. Lemma2 should point to A.3. We have updated the paper. A.2 is included to explain in detail the process of how we obtain the derivative of the Lagrangian.
> 6. We have updated A.3 to provide more detailed derivation. We apologize for the reference error. It should be ”(28) is an application of (7)” in the previously submitted version. Please note that the numbering of equations has been changed in the latest version.
> 7. The source code of MOTO[I] is not available. Besides, we consider GAC[II] as a more powerful version of MOTO, a more competitive baseline for us to compare with. According to the authors' claim in their GAC paper --  ”MOTO learns a sequence of time-dependent log-linear Gaussian policy and a time-dependent critic, while GAC learns a log-nonlinear Gaussian and a more complex critic”.
> Regarding the comparison with GAC, we only completed 2 testing tasks for GAC before the submission deadline due to the limited computation resources we had and the fact that GAC is still extremely computationally expensive. We have completed the testing of GAC on all six tasks and updated our comparison results in Fig.1 in the revised paper.
> 8. Following the suggestion by the reviewer, we have revised the use of words to make them more accurate. We are also testing GAC with more trails (doubling the number of random seeds). The execution of additional experiments has not completed yet due to the limited computation resources we have. For now, the available results are consistent with our previous observation/conclusion. We will update the paper with the complete results as soon as they come out.
> Besides, in Fig.1, we set $\eta$ according to its action space of each task (suggested by SAC) and we simply set $\tau=0.005$ for all tasks. In fact, for each task, $\tau$ should be tuned to get optimized results. We updated Appendix E with Fig.4 to show the experimental results with the tuned $\tau$ for each task. The improvements in three tasks (Humanoid, Swimmer, and Ant) are much more evident.
>
> [I]: Riad Akrour, Abbas Abdolmaleki, Hany Abdulsamad, Jan Peters, Gerhard Neumann. Model-Free Trajectory-based Policy Optimization with Monotonic Improvement. J. Mach. Learn. Res. 19: 14:1-14:25 (2018)
>
> [II]: Voot Tangkaratt, Abbas Abdolmaleki, Masashi Sugiyama. Guide Actor-Critic for Continuous Control. ICLR (Poster) 2018

---

> > ### Comment · AnonReviewer4 · 2020-11-22
> > **Response to Response**
> >
> > Point 1. It looks like this still isn't available yet
> >
> > Point 2. The third constraint still appears to be an expectation over nothing
> >
> > Point 4. Based on your description, the old policy should be one of the things given in the lemma statement.  There is nothing in the lemma defining it.  Also, it would be more natural to state the last part of the result as "If" rather than "Assume that"
> >
> > Point 5. If you are going to keep the material is A.2 then something in the main text should point to it.
> >
> > Point 6. I still do not understand the revised derivation, possibly because there is a reliance on techniques from prior work that are not clearly explained.  Right at the start of the proof I have a problem because the lemma statement defines the new \pi, but the start of the proof provides a different definition.  Why are (29) and (30) equivalent?  The main text talks a bit about this formulation and the representation issue, but is unclear about the precise technical details.  Then there is still a ... at the tail end of the proof.
> >
> > Point 8. The claims in the experimental section are better, but elsewhere (e.g. abstract and conclusion) I still think they are too strong.  I'm also not necessarily convinced by the results showing that tuning \tau leads to improved performance over the baselines because as far as I can tell they have not had the benefits of similar tuning.

---

> > > ### Author Response · Authors · 2020-11-25
> > > **Further response**
> > >
> > > We thank the reviewer for all feedback which help us to further revise and improve our paper!
> > >
> > > Point 1. We apologize for missing the inclusion of similar work in the Introduction. A new paragraph is not included in the Introduction to highlight MOTO, GAC, and TRUST-PCL are existing work on unifying the trust region constraint and the entropy constraint, and our motivation is to address their problem of inefficiency.
> > >
> > > Point 2.  We agree with the reviewer that $\int_{\mathcal{S} \times A} \rho_{\pi}(s) \pi(a|s) ds da=1$ should be the accurate form to describe the constraint. We replaced the third constraint with $\mathbb{E}_{{\rho^{\pi}(s)}}\int \pi(a|s)da = 1$ to make it consistent with each others.
> > >
> > > Point 4. The old policy is the condition of the lemma statement. We have revised Lemma 2 to be "Given a policy $\pi$ and an old policy $\hat{\pi}$, define a new policy ...". We are now using "If" instead of "Assume that" to claim conditions. Thanks!
> > >
> > > Point 5.  Eq.10 should refer to A.2 for checking details. We have updated the paper to reflect this.
> > >
> > > Point 6. We have revised Lemma 2 in the paper. In order to explain the derivation more concisely and more intuitively, we provided an alternative derivation to prove policy improvement. We have updated derivation in A.3.
> > >
> > > Point 8. Thanks! We have now revised the wording in the abstract and the conclusion. As $\tau$ represents the desired maximum KL-divergence, an optimized $\tau$ can help to tame the variations during the policy exploration. In fact, the maximum of KL divergence among different tasks can be very different. The hyperparameter analysis study in [1] and the continuous tasks evaluated in [2] have both shown the benefit of setting the suitable $\tau$ for different tasks.
> > > The experiments with tuned $\tau$ confirmed its usefulness. The results in Fig.4 show that the selection of proper $\tau$ value for each task can notably boost the performance of TEAC.
> > > Regarding our previous reply, we have finished the experiments of doubling the number of the random seeds in Fig.1. The results are depicted in Fig.5, which are consistent with the performances/observations shown in Fig.1. Please note that in both Fig.1 and Fig.5, we simply set $\tau=0.005$ for all tasks. In fact, for each task, $\tau$ should be tuned to get optimized results. Fig.4 has shown the experimental results with the tuned $\tau$ for each task. The improvements in three tasks (Humanoid, Swimmer, and Ant) are much more evident.
> > >
> > > [1]: Ofir Nachum, Mohammad Norouzi, Kelvin Xu, Dale Schuurmans. Trust-PCL: An Off-Policy Trust Region Method for Continuous Control. CoRR abs/1707.01891 (2017)
> > >
> > > [2]: Joni Pajarinen, Hong Linh Thai, Riad Akrour, Jan Peters, Gerhard Neumann. Compatible Natural Gradient Policy Search. CoRR abs/1902.02823 (2019)

---

### Official Review · AnonReviewer1 · 2020-11-11
**An interesting idea with several technical issues**

**Rating:** 5
**Confidence:** 4

**Review:**

This paper proposes Trust Entropy Actor Critic (TEAC), a novel algorithm for reinforcement learning (RL) combining the idea of TRPO/PPO and max-entropy RL, together with the corresponding critic, actor and dual updates. The high level idea is that trust region methods ensure stability by constraining the KL divergence from the previous policy, while entropy regularization encourages exploration, and hence combining the two may achieve the best of both worlds and obtain a good trade-off between stability and exploration. To achieve this goal, the authors propose to augment the original trust-region subproblem in TRPO with an additional constraint on the lower bound of the policy entropy (together with two other trivial constraints corresponding to the validity of the policy in the MDP framework). Then by forming the Lagrangian function and setting the gradient to zero, the authors obtain both the critic (value) and actor (policy) updates with different choices of dual variables (corresponding to the two trivial constraints), together with the dual updates. Numerical experiments compared to some popular baseline RL algorithms are also reported to demonstrate the improvement of TEAC compared to the existing works.

in general, the high level idea is interesting and reasonable, and some empirical improvement is also shown. However, there are several undefined terms and technical issues/mistakes that largely downgrade the quality and contribution of this paper.
1. $Q^{\pi}$ in (1) and (2) is not defined, and the meaning of $E_{\rho_{\pi}(s),\pi(a|s)}$ is unclear. Also, according to standard literature like [1, Chapter 4], the expected reward $\mathcal{J}(\pi)$ should indeed be something like $\frac{1}{1-\gamma}E_{s\sim \rho_{\pi}, a\sim \pi(\cdot|s)}r(s,a)$, so the term inside the expectation should probably be the reward $r(s,a)$ instead of the Q function $Q^{\pi}(s,a)$.
2. The formulation in (2) is confusing. Firstly, it is unclear what the notation $E_{\rho_{\pi}(s)\pi(a|s)}=1$ means, and what the difference is between $E_{\rho_{\pi}(s)\pi(a|s)}$ and $E_{\rho_{\pi}(s),\pi(a|s)}$. Similarly, it is unclear what $V$ is (as a function of $\pi$ or a free variable), and what the notation $E_{\rho_{\pi}(s)\pi(a|s)p(s’|s,a)}$ means. In general, all the terms appearing in the optimization problem should either be some constants or a function of $\pi$, but this is not made clear by the authors. Also, it’s unclear why the authors switch the orders of $\pi$ and $\pi_{\rm old}$ in the KL divergence compared to the original TRPO formulation, and some explanations should be provided.
3. The formula (4) seems to be weird. Firstly, according to the right-hand side, it seems that the left-hand side should indeed be the (s,a)-entry of the gradient instead of the entire gradient. Secondly, it is unclear why $\rho(s)$ disappears in the second equality.
4. The authors argue that the dual variables $\nu$ and $\lambda$ can be taken arbitrarily as the policy is parametrized as a Gaussian via neural networks so that it is always a valid randomized policy (satisfying the third constraint in (2)), while the “value function” always satisfies the Bellman equation corresponding to the fourth constraint in (2). However, with the neural network parametrization $\theta$, the optimization problem should also be solved w.r.t. $\theta$ instead of $\pi$, in which case (4) no longer makes sense (since it’s differentiating w.r.t. $\pi$ instead of $\theta$). Also, as mentioned above, the value function $V$ is undefined and hence the second claim does not make much sense. In addition, if the fourth constraint in (2) corresponds to the Bellman equation, then it is weird why no reward $r$ or discount factor $\gamma$ is involved there.
5. In Lemma 1, the “trust entropy Q-value” is undefined.
6. In (10), why can $\exp(-(\alpha+\beta+\lambda)/(\alpha+\beta))$ serve as a normalization term? In particular, why does it hold that $\int_a\pi_{\rm old}(a|s)^{\alpha/(\alpha+\beta)}\exp(Q^{\pi}(s,a)/(\alpha+\beta))da$ always equal to $\exp(-(\alpha+\beta+\lambda)/(\alpha+\beta))$ for any state $s$?

Also, the experimental part has some limitations.
1. The numerical improvement is not very convincing, as it seems that TEAC only improves over SAC in two out of six tasks and only has significant improvement on one of these two tasks. Some more thorough comparisons are needed to validate the empirical advantage of the proposed method.
2. The actually implemented algorithm, Algorithm 1 contains several tricks unexplained in the main text. In particular, why do we need two target critic networks? And in the update (38), should $\bar{\phi}$ be $\bar{\phi}_i$?

Finally, please find some miscellaneous suggestions/comments on additional references, technical issues fixing and typo fixing below.
1. The work [2] seems to be closely related to the high level idea of this paper, and should better be compared with.
2. In the abstract, “transforms” should be “transform”. In the first paragraph of the introduction, “learning process” should be “learning processes”.
3. In the last paragraph of the introduction, “guaranteed to converge” is not very accurate. In fact, only the critic/policy evaluation is guaranteed to converge, and the authors show that policy improvement does hold (but may or may not lead to eventual convergence of the whole TEAC algorithm).
4. In the first paragraph of Section 2, the definition of $\rho_{\pi}$ is not provided, although from the literature it probably means the discounted state-visitation distribution/measure. The authors should provide a clear definition for self-contained-ness, and the term “state of the trajectory distribution” does not seem to make much sense and should better be replaced with more standard terminologies like “discounted state-visitation distribution/measure”. Also, “qantified” should be “quantified”.
5. In TEAC, the parameters $\tau$ and $\eta$ are always fixed. However, as the algorithm proceeds, it may not make much sense to keep a constant exploration power as required by the constant $\eta>0$. Will a decreasing $\eta$ be a better choice? Some discussions on this should better be provided.
6. In (3), $\rho(s)$ should be $\rho_{\pi}(s)$, and the dual variables $\alpha$ and $\beta$ should be required to be non-negative. Accordingly, the dual updates should probably better be projected onto the non-negative orthant.
7. In the sentence before (4), “derivation” should be “derivative”.
8. In (5), $Q$ should be $Q^{\pi}$.
9. In (20), the entropy terms lack right parentheses.

Hence in general, I think the paper is still not ready for publication and needs substantial fixing and improvement.

[1] Agarwal, Alekh, Nan Jiang, and Sham M. Kakade. Reinforcement learning: Theory and algorithms. Technical Report, Department of Computer Science, University of Washington, 2019.

[2] Pajarinen, Joni, Hong Linh Thai, Riad Akrour, Jan Peters, and Gerhard Neumann. "Compatible natural gradient policy search." Machine Learning 108, no. 8-9 (2019): 1443-1466.

---

> ### Author Response · Authors · 2020-11-15
> **Response to Reviewer #1 (part 1/2)**
>
> We thank the reviewer for the very detailed and constructive feedback! We have updated the paper and highlighted the corrections/detailed illustrations following the reviewer's comments.
> 1. We have now updated the paper and defined $Q^\pi$ and $V^\pi$ in Section 2. Accordingly, we revised Eq.1 of $\mathcal{J}(\pi)$. It should be with respect to $\rho_{\pi}(s)\pi(a|s)$, instead of $\rho_{\pi}(s), \pi(a|s)$, clarifying that the pair $(s,a)$ is in fact drawn from the state-action marginal of the trajectory distribution induced by policy $\pi(a|s)$. Throughout the paper, all ”$\rho_{\pi}(s), \pi(a|s)$” have been replaced by $\rho_{\pi}(s)\pi(a|s)$. We thank the reviewer for pointing this out.
> 2. In Eq.2, the third constraint ensures that the state-action marginal of the trajectory distribution is a proper probability density function. Regarding the fourth constraint, as the state marginal of the trajectory distribution needs to comply with the policy $\pi(a|s)$ and the system dynamics $p(s'|s, a)$, i.e., $\rho_{\pi}(s) \pi(a | s) p(s'|s, a) = \rho_{\pi}(s')$, meanwhile the direct matching of the state probabilities is not feasible in continuous state spaces, the use of $V^{\pi}(s')$, which can be also considered as state features, helps to focus on matching the feature averages. We have updated the descriptions regarding these two constraints in the paper.
> The difference between our KL divergence and the original TRPO formulation can be attributed to forward-KL and reverse-KL. Both TRPO and TEAC aim to find the optimal policy. However, TRPO, without the entropy term constraint to encourage the exploration, utilizes the forward-KL which is known as ”Mean-Seeking” to retain the average ”action modes” of the old policy. In comparison, TEAC, with the entropy constraint to encourage the exploration, uses the reverse-KL, known as ”Mode-Seeking”, as a better fit because KL-divergence serves as imposing a stricter restriction of the policy updates. It is a common practice to apply the reverse-KL to guide the policy improvement in consideration of the entropy constraints, such as [I],[II],[III], following REPS [IV].
> 3. Eq.4 is derived by taking the derivative of $\mathcal{L}$ with respect to $\pi$. For example, the first term in the right-hand side of Eq.3 can be rewritten as $E_{\rho(s)\pi(a|s)}[Q^{\pi}(s,a)] = E_{\rho(s)}[\int \pi(a|s) Q^{\pi}(s,a) da]$ , and its derivative w.r.t $\pi$ is $\mathbb{E}_{\rho(s)}[\int Q^{\pi}(s,a) da]$.
> There are two ways to derive the second line of Eq.4. 1) Since $\alpha+\beta+\lambda$ is a constant, we can move the term into the expectation and integral. Because we can add bounded constant to the Q function without changing the solution of Eq.2, we assume that Q is shifted so that the term inside the integral is non-negative. In this case, the expectation equals zero if and only if the term inside is zero (Eq.4). 2) Consider the derivative is with respect to $\pi(a|s)$ for each pair $(s,a)$. In this case, given a specific pair of $(s, a)$, the derivative is zero for other states and actions. We will add the explanation to the paper.
> 4. Eq.4 is not the objective to be optimized. Instead, Eq.4 is used to define our new bellman equation. With the new bellman equation (Eq.6 and Eq.7), the optimization problem will be solved w.r.t optimizing the parameters $\theta$, $\phi$, $\alpha$ and $\beta$. Thanks for the suggestion, and we have formally defined Q and V in Section 2.
> 5. We have clarified the ”trust entropy Q-value” on Page 4. The Q function obtained by iteratively applying the modified Bellman backup operator in Eq.6 and Eq.7 is the trust entropy Q-value in our framework. As with the standard Q-function, value function, we can relate the trust entropy Q-value to the trust entropy state-value at a future state via a modified Bellman equation, where the trust entropy state-value is derived from Eq.5.
> 6. The normalization term can be derived through the third constraint in Eq.2. We have added more detailed descriptions about it in appendix A.2. (Page 12)
>
>
> [I]: Riad Akrour, Abbas Abdolmaleki, Hany Abdulsamad, Jan Peters, Gerhard Neumann. Model-Free Trajectory-based Policy Optimization with Monotonic Improvement. J. Mach. Learn. Res. 19: 14:1-14:25 (2018)
>
> [II]: Ofir Nachum, Mohammad Norouzi, Kelvin Xu, Dale Schuurmans. Trust-PCL: An Off-Policy Trust Region Method for Continuous Control. ICLR (Poster) 2018
>
> [III]: Voot Tangkaratt, Abbas Abdolmaleki, Masashi Sugiyama. Guide Actor-Critic for Continuous Control. ICLR (Poster) 2018
>
> [IV]: Jan Peters, Katharina Mülling, Yasemin Altun. Relative Entropy Policy Search. AAAI 2010

---

> > ### Author Response · Authors · 2020-11-15
> > **Response to Reviewer #1 (part 2/2)**
> >
> >
> > The experimental part:
> >
> > 1. $\eta$ and $\tau$ are the hyperparameters in our framework. In Fig.1,  we set $\eta$ according to its action space of each task (suggested by SAC) and we simply set $\tau=0.005$ for all tasks. In fact, for each task, $\tau$ should be tuned to get optimized results. We updated Appendix E with Fig.4 to show the experimental results with the tuned $\tau$ for each task. The improvements in three tasks (Humanoid, Swimmer, and Ant) are much more evident.
> > 2. Following the settings of TD3[I] and SAC[II], the use of two critic networks in Algorithm 1 is to mitigate the overestimation bias in the policy improvement step. We have updated the description in the paper. The reviewer is right that (38) should be $\bar{\phi}_i \leftarrow \kappa \phi_i+(1-\kappa) \bar{\phi}_i$ in the previous version. In the latest version, to clarify the difference between the sample number and the critic network number, we use ”j” to replace ”i” in previous (37) and (38) (which are Eq.43 and Eq.44 now).
> >
> > Other suggestions:
> >
> > 1. We have tried to include [2] in the experiment comparison. However, its source code has not been publicly available. We have emailed the authors for the code, but have not received any reply yet. In work [2], the parameters of the approximation of the Monte-Carlo estimates are estimated as a natural gradient using the conjugate gradient method, which requires multi-steps to compute in each update procedure. Besides, [2] needs to integrate over actions, similar to GAC, which causes the difficulty of dealing with high-dimensional action space (much more computation is required). Hence, we believe that our work holds a competitive advantage over [2]. We would like to verify our conjectures and observations when their code is available.
> > 2. Many thanks! We have fixed all these typos.
> > 3. In the last paragraph of the introduction, we meant to say that we have proved that the critic/policy evaluation is converged and the policy improvement can be guaranteed to be converged in Actor Side. We are sorry to cause the confusion of guaranteed convergence of TEAC. However, empirically, TEAC converged for most tasks, as shown in Fig.1.
> > 4. Thanks for the suggestion! We have clarified that $\rho_{\pi}$ is the discounted state distribution on Page 2.
> > 5. In the main manuscript, we followed SAC to set $\eta$ as the negative of the action space dimension in TEAC. As a hyperparameter in TEAC, there is no requirement that $\eta>0$. We agree with the reviewer that decreasing $\eta$ along with the training progress would be a good choice. In Appendix, we did provide the comparison with different heuristics of $\eta$'s settings suggested in the literature, including 1) setting $\eta$ as the negative of action space dimension (SAC), 2) the decreasing version (Eq.39) as in MORE, and 3) a GAC setting (Eq.40). However, the difference regarding the performance is not noticeable (Fig.3 in Appendix B). It is probably because that the negative action space is usually less than $E_0$ which is the entropy of the randomized policy drawn from Gaussian distribution at the beginning and the starting entropy threshold in Eq.39 for most cases. The discussion of $\tau$ can be found in the answer to the first question in the experimental part.
> >
> > 6. Yes, $\rho(s)$ should be $\rho_\pi(s)$. For the sake of brevity, we use $\rho(s)$ to represent $\rho_\pi(s)$ starting from Eq.3. We have updated the paper to clarify this. And the reviewer is correct that the dual variables are non-negative and should be projected onto the non-negative orthant. In the implementation (in the source code of TEAC and SAC), we update the log form of the Lagrangian variables to enable the updates free of the non-negative constraints and then take the exponential function to project them in the non-negative space when computing the loss functions of actor and critic.
> > 7-9. Thank you very much for the detailed comments! we have updated the paper to address all of them.
> >
> > [I]: Scott Fujimoto, Herke van Hoof, David Meger. Addressing Function Approximation Error in Actor-Critic Methods. ICML 2018: 1582-1591
> >
> > [II]: Tuomas Haarnoja, Aurick Zhou, Pieter Abbeel, Sergey Levine. Soft Actor-Critic: Off-Policy Maximum Entropy Deep Reinforcement Learning with a Stochastic Actor. ICML 2018: 1856-1865
> >
> > [2]: Pajarinen, Joni, Hong Linh Thai, Riad Akrour, Jan Peters, and Gerhard Neumann. Compatible natural gradient policy search. Machine Learning 108, no. 8-9 (2019): 1443-1466.

---

> > ### Comment · AnonReviewer1 · 2020-11-22
> > **Response to response (part 1/2)**
> >
> > 1. Thanks for the correction. However, I think $\rho_{\pi}$ should be $p_0$ instead, given that $\mathcal{J}(\pi)=\mathbb{E}_{s\sim p_0}V^{\pi}(s)$, if I understand correctly. If this is the case, all subsequent appearances of $\rho_{\pi}$ should also be replaced with $p_0$.
> > 2. it might be better to simply write the third constraint as $\int_{\mathcal{S}\times\mathcal{A}}\rho_{\pi}(s)\pi(a|s)dsda=1$. Otherwise, it is still unclear to what quantity is the expectation operator applied. And I appreciate the authors explanations on the forward and reverse RL.
> > 3. I don’t think the derivations are valid here. For the first argument, since $Q^{\pi}$ is also a function of $\pi$, it seems that the authors forget about the derivative of $Q^\pi$ w.r.t. $\pi$. For the second argument, the derivative has already been taken in the first step and the issue here is that now the expectation over $\rho(s)$ disappears, which I still don’t think makes sense.
> > 4. Thanks for some clarifications, but this still does not explain why the authors can choose $\nu$ and $\lambda$ arbitrarily, as pointed out in my original review.
> > 5. If the ”trust entropy Q-value” is directly defined as “obtained by iteratively applying the modified Bellman backup operator”, then it is unclear what it stands for and in particular what the relationship is between this trust entropy Q-value and the original Q-value. And since the trust entropy state-value (V) is defined by the trust entropy Q-value, it also does not give any explanation on what the trust entropy Q-value is indeed.
> > 6. The proof seems wrong. The authors implicitly assumed that equation (24), followed from setting $\nu=0$, satisfies the constraints in (2) exactly (especially the third constraint). However, as mentioned in item 4 above, I don’t think this argument is valid. It is only valid if the authors are directly working with the Gaussian parametrized policies and taking derivatives w.r.t $\theta$. When working with the policy $\pi$ itself, such constraints will not be automatically enforced. See the discussion after Remark 5.1 of this paper: https://arxiv.org/pdf/1908.00261.pdf for a more formal explanation.

---

> > > ### Author Response · Authors · 2020-11-25
> > > **Further response**
> > >
> > > We thank the reviewer for all feedback which help us to further revise and improve our paper!
> > >
> > > 1. Thanks for the suggestion! We use $\rho_\pi$ in the paper to implicate that $\rho_\pi$ is the stationary distribution of states under $\pi$ and independent of $s_0$ for all policies. This follows the convention in Prof. Sutton's 1999 NIPS work - policy gradient methods for RL with function approximation. We have further updated the illustration in the paper.
> > > 2. We agree with the reviewer that $\int_{\mathcal{S} \times A} \rho_{\pi}(s) \pi(a|s) ds da=1$ should be the accurate form to describe the constraint. We replaced the third constraint with $\mathbb{E}_{{\rho^{\pi}(s)}}\int \pi(a|s)da = 1$ to make it consistent with each others.
> > > 3. The reviewer is right. The previous denotation is not correct. The argument "$Q^{\pi}(s,a)$" should be replaced by "$\hat{Q}(s,a)$" which represents that "$\hat{Q}(s,a)$" is a critic parameterized by neural networks, instead of a function of policy $\pi$. Then, the derivative of $\mathcal{L}$ w.r.t $\pi$ does not involve the derivative of $Q$ w.r.t. $\pi$. The paper has been updated accordingly.
> > > After the change to $\hat{Q}(s,a)$, it might be helpful to understand Eq.4 by switching the second line and third line in Eq.4. Given the first line is zero, the second line being zero can be considered from two aspects: 1) Since $\alpha+\beta+\lambda$ is a constant, we can move the term into the expectation and integral. Because we can add bounded constant to the Q function without changing the solution of Eq.2, we assume that Q is shifted so that the term inside the integral is non-negative. In this case, the expectation equals zero if and only if the term inside is zero. Hence the expectation equals zero and the term inside the integral (which is the second equality) equals zero, which makes the first equality equals the second equality. 2) Consider the derivative is with respect to $\pi(a|s)$ for each pair $(s,a)$. In this case, given a specific pair $(s, a)$, the derivative is zero for other states and actions. Hence we can remove the expectation and the integral.
> > > 4. We apologize that we misunderstood the reviewer's previous comment regarding $\nu$ and $\lambda$. For $\lambda$, the last constant term in Eq.5 can be ignored as it does not affect the Bellman iteration when we use neural networks to approximate the function. Thus, the $\lambda$ can be arbitrary. For $\nu$, as "the direct matching of the state probabilities is not feasible in continuous state spaces, the use of $\hat{V}(s')$ in the fourth constraint, which can be also considered as state features, helps to focus on matching the feature averages." Besides $\hat{V}(s')$, we could utilize any form of state features. Thus, $\nu \hat{V}(s')$ can be seen as another form of state features. Therefore, $\nu$ can be arbitrary. We have revised the paper regarding the explanation now.
> > > 5. We have updated the paper to make the description clearer. For any state-action value function $Q:S\times A\rightarrow \mathbb{R}$, we iteratively apply the modified Bellman backup operator (Eq.6 and Eq.7) to compute the trust entropy Q-value, which is represented as $Q(s,a)$, in our framework.
> > > 6. Thanks. Appendix A.2 is not meant to prove the global convergence of the policy improvement. The purpose of A.2 is to explain the closed-form solution of the policy. The reviewer is correct that the global convergence can be guaranteed only when we apply the Gaussian parameterized policies and use gradient descent method to update the policies. We have acknowledged this point in Section 3.3: "With the assumption of policy being Gaussian, the policy improvement can be guaranteed in our framework." Now we have updated the proof of guaranteed policy improvement stressing a utilized parameterized Gaussian policy.

---

### Decision · Program_Chairs · 2021-01-07
**Final Decision**

**Decision:**

Reject

**Comment:**

The paper proposes a reinforcement learning algorithm that combines trust region policy optimization and entropy maximization. The starting point is the Lagrangian of a constrained optimization problem that upper bounds the change in the policy and lower bounds the entropy of the policy. The paper proves that the algorithm converges, and evaluates it experimentally in MuJoCo domains.

The main issues raised by the reviewers were related to the proofs (see especially R1) and experimental evaluation (R4). The authors did a great job improving the paper during the discussion phase, but some of the issues remain unresolved, and thus reviewers find the paper not to be ready for publication. Thus, I'm recommending rejection.